# Chiral self-sorted multifunctional supramolecular biocoordination polymers and their applications in sensors

Xiaobo Shang[1,7], Inho Song [1,7], Gwan Yeong Jung[2], Wanuk Choi[3], Hiroyoshi Ohtsu [4], Jeong Hyeon Lee[2], Jin Young Koo[5], Bo Liu[6], Jaeyong Ahn[1,7], Masaki Kawano[4], Sang Kyu Kwak [2] & Joon Hak Oh [1,7]

Chiral supramolecules have great potential for use in chiral recognition, sensing, and catalysis. Particularly, chiral supramolecular biocoordination polymers (SBCPs) provide a versatile platform for characterizing biorelated processes such as chirality transcription. Here, we selectively synthesize homochiral and heterochiral SBCPs, composed of chiral naphthalene diimide ligands and Zn ions, from enantiomeric and mixed $R$-ligands and $S$-ligands, respectively. Notably, we find that the chiral self-sorted SBCPs exhibit multifunctional properties, including photochromic, photoluminescent, photoconductive, and chemiresistive characteristics, thus can be used for various sensors. Specifically, these materials can be used for detecting hazardous amine materials due to the electron transfer from the amine to the SBCP surface and for enantioselectively sensing a chiral species naproxen due to the different binding energies with regard to their chirality. These results provide guidelines for the synthesis of chiral SBCPs and demonstrate their versatility and feasibility for use in various sensors covering photoactive, chemiresistive, and chiral sensors.

[1] Department of Chemical Engineering, Pohang University of Science and Technology (POSTECH), Pohang, Gyeongbuk 37673, Republic of Korea. [2] School of Energy and Chemical Engineering, Ulsan National Institute of Science and Technology (UNIST), UNIST-gil 50, Ulsan 44919, Republic of Korea. [3] Center for Ordered Nanoporous Materials Synthesis, School of Environmental Science and Engineering (POSTECH), Pohang, Gyeongbuk 37673, Republic of Korea. [4] Department of Chemistry, School of Science, Tokyo Institute of Technology, 2-12-1 Ookayama, Meguro-ku, Tokyo 152-8550, Japan. [5] Department of Chemistry, Pohang University of Science and Technology (POSTECH), Pohang, Gyeongbuk 37673, Republic of Korea. [6] Department of Chemistry, Zhejiang University, 310027 Hangzhou, China. [7] Present address: School of Chemical and Biological Engineering, Institute of Chemical Processes, Seoul National University, 1, Gwanak-ro, Gwanak-gu, Seoul 08826, Republic of Korea. These authors contributed equally: Xiaobo Shang, Inho Song. Correspondence and requests for materials should be addressed to S.K.K. (email: skkwak@unist.ac.kr) or to J.H.O. (email: joonhoh@snu.ac.kr)

Chirality is widespread in nature and common in the most abundant forms of matter[1]. Self-assembly and self-organization to create higher-order functional structures play an important role in natural systems[2]. Chiral recognition is important in biological processes and organic synthesis[3]. Compared with homochiral assemblies, heterochiral systems are unique and largely dependent on the mixing of chiral enantiomers[1]. Chiral recognition between two homochiral pairs of the heterochiral system can lead to self-recognition or self-discrimination. In most cases, homochiral assemblies via self-recognition are more commonly formed from a mixed chiral system rather than heterochiral assemblies via self-discrimination. However, in some cases, heterochiral assemblies are spontaneously formed during phase separation in heterochiral systems; this is not common since the energy difference between enantiomers is relatively small[4]. Recently, supramolecular chirality has attracted great attention due to its promising applications in chiral recognition, sensing, and catalysis[1]. Several optical devices based on the supramolecular chirality and other biological applications have also been reported[1,5,6].

Chiral coordination polymers (CPs) show diverse structures and topologies with many potential applications[7–10]. In general, they can be synthesized by introducing chiral ligands, chiral templates, chiral physical environments, or through spontaneous resolution from achiral materials without chiral auxiliaries[7–10]. To produce chiral supramolecular biocoordination polymers (SBCPs), which exhibit unique biological characteristics and represent an excellent platform for understanding biorelated processes, amino acids are a particularly promising monomeric unit due to their remarkable molecular recognition capability, controllable chirality, and distinctive sequence-specific self-assembling properties[11]. Based on these characteristics, various molecules with amino acides were investigated on their molecular assemblies and exploitations for advanced functionalities[12].

Naphthalene diimides (NDIs) possess high electron affinities, high charge carrier mobility, and excellent thermal and oxidative stabilities, making them promising candidates for various applications such as organic field-effect transistors and organic photovoltaic devices[13,14]. In addition, NDIs are chemically robust aromatic planes that can produce stable CPs due to their strong π–π interactions among neighboring molecules. The use of NDIs in coordination complexes has been well documented;[15,16] however, few studies have explored NDI-based biocoordination polymers (BCPs)[17–19].

In recent years, there has been increasing interest in the use of CPs as next-generation functional materials in electronics and photonics[20,21]. Owing to their high surface area and robust chemical tunability based on a "bottom-up" synthetic approach, CPs have been used in sensors[22–24]. CP-based sensors for detecting volatile organic compounds (VOCs) can be classified into four major categories: luminescent sensors, electrochemical sensors, electromechanical sensors, and miscellaneous sensors[25]. Typically, CPs do not possess electrical conductivity due to the insulating characteristics of organic ligands. However, their crystalline nature has the ability to form composites with other electrically active components. These sensors require adsorption of the analyte on the surface of the materials. Owing to their large surface area, one can observe enhanced diffusion of chemical species into and out of CPs. However, few studies have explored the sensor applications of chiral CPs. On the other hand, photochromic materials have attracted considerable interest due to their applications in information storage, optical switches, and photomechanics[26]. Although numerous photochromic families have been reported, those based on electron transfer chemical processes (especially for CPs) are relatively few[27]. Another approach to optimizing the performance of CPs is the use of light

to change their properties. CPs with photoswitchable properties and modifiable framework features through a reversible approach by means of external light stimuli have also been reported[28]. However, the research on CP-based photoswitching is still in its infancy[29]. To the best of our knowledge, few studies have attempted to tune the photoswitching properties of SBCPs by manipulating the redox ligand NDI.

In this study, we introduce amino-acid functionalized NDIs as organic linkers to construct multifunctional homochiral and heterochiral SBCPs (i.e., AlaNDI-Zn; composed of a NDI ligand with alanine termini (AlaNDI) and Zn ions) and use them as a platform to characterize chirality transcription of amino acids in BCPs. Unlike previous photoswitchable CPs that use reversible structural or geometrical changes of the ligands upon light irradiation[28], the photochromic and photoswitchable SBCPs developed herein have redox-active NDI ligands. Such redox mechanism can be attributed to the formation of radicals in SBCPs upon exposure to UV light. The devices based on the SBCPs exhibit outstanding photoresponsivity and detectivity in UV spectral region. In addition, the SBCPs have been applied to photoluminescence (PL) sensing of a trace amount of the harmful chemical hydrazine with the detection limit of $3.2 \times 10^{-6}$ M, which is comparable to the previous report[30]. This simple and feasible method can be used to detect a low concentration of electron-rich amines. Interestingly, our homochiral SBCP can enantioselectively detect chiral naproxen by fluorescence quenching with high sensitivity because of their different binding strengths. Furthermore, we have used insulating SBCPs for chemiresistive sensing of electron-rich VOCs, such as methanol, ethanol, and aniline with enhanced conductivity due to the reduction of energy gap. The detection limit for electron-donating aniline reaches 16 ppm with the insulative SBCPs, which is comparable to the reported high-performance chemiresistive sensing for ammonia or amines using conductive CPs[22,23]. The NDI-based SBCPs developed herein have great potential to be used as multifunctional sensors covering photoactive, chemiresistive, and chiral sensing.

## Results

**Synthesis of ligands and AlaNDI-Zn SBCPs.** $H_2$AlaNDI homochiral ligands were synthesized from 1,4,5,8-naphthalene-tetracarboxylic dianhydride (NTCDA) and L-alanine or D-alanine, which were refluxed for 12 h in pyridine[31], and AlaNDI-Zn SBCPs were synthesized using a solvothermal reaction between the ligands and zinc iodide in N,N-dimethylformamide (DMF) in a sealed stainless-steel tube with a Teflon liner at 120 °C for 72 h with yields of 29, 27, and 21% for (S)-AlaNDI-Zn, (R)-AlaNDI-Zn, and (Rac)-AlaNDI-Zn, respectively.

**Characterization of ligands and AlaNDI-Zn SBCPs.** The morphologies of synthesized homochiral and heterochiral SBCPs were observed using scanning electron microscopy (Supplementary Fig. 1). Homochiral and heterochiral SBCPs showed similar ribbon-shaped morphologies with a length of 20–100 μm, width of 10–60 μm, and thickness of 1–10 μm. Single-crystal X-ray diffraction (XRD) analysis revealed that (S)-AlaNDI-Zn and (R)-AlaNDI-Zn crystallize as a monoclinic $P2_1$ space group from pure enantiomers. On the other hand, (Rac)-AlaNDI-Zn crystallizes as a monoclinic $P2_1$ space group and consists of 1D chains in which (R)-ligands and (S)-ligands are aligned alternatively (Fig. 1a–c). In all crystal structures, zinc ions have four-coordinated tetrahedral geometries and are coordinated with two oxygen atoms originating from the carboxylate group of AlaNDI ligands and two oxygen atoms arising from ligated DMF. The two carboxylate groups in each AlaNDI ligand are arranged

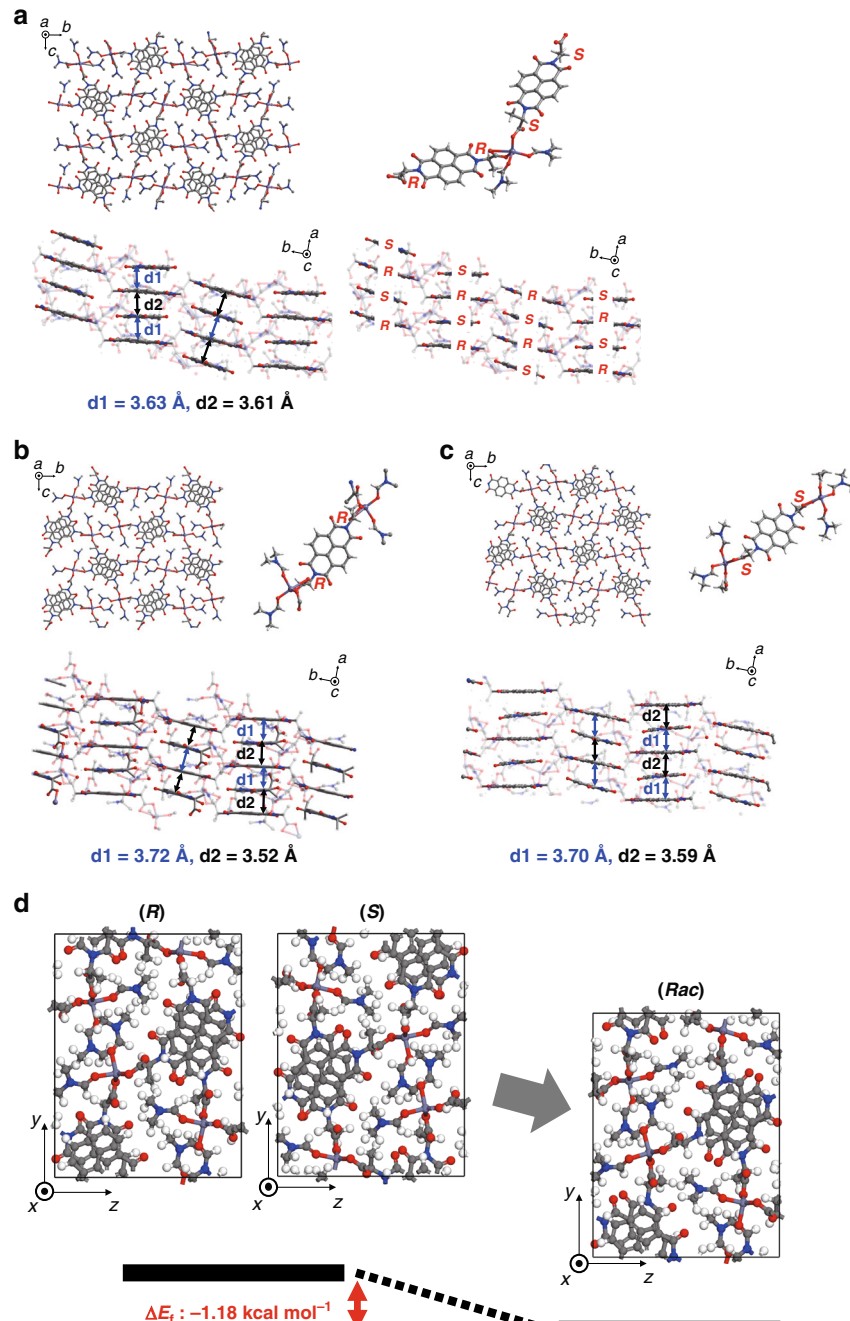

**Fig. 1** Crystal structures of homochiral and heterochiral AlaNDI-Zn SBCPs and their chiral self-discrimination phenomena. **a**–**c** Crystal structures of **a** heterochiral (*Rac*)-AlaNDI-Zn SBCP (top-left shows the *a*-axis projection, top-right shows a unit description containing two AlaNDI units in a 1D chain with the chirality description, bottom-left shows a π–π stacking scheme with centroid–centroid distances, and the bottom-right shows the chiral configuration in this SBCP), **b** homochiral (*R*)-AlaNDI-Zn SBCP (top-left shows the *a*-axis projection, top-right shows a unit description containing one AlaNDI unit in a 1D chain with the chirality description, and bottom shows a π–π stacking scheme with centroid–centroid distances), and **c** (*S*)-AlaNDI-Zn SBCP (top-left shows the *a*-axis projection, top-right shows a unit description containing one AlaNDI unit in a 1D chain with the chirality description, and bottom shows a π–π stacking scheme with centroid–centroid distances). Atom coloring: Zn, thin-purple, C, gray, N, blue, O, red, and H, white. Except for the top-right figure in each, hydrogen atoms were omitted for clarity. **d** Formation energy (ΔE_f) calculation results for chiral discrimination phenomena in heterochiral SBCPs. The carbon, hydrogen, oxygen, nitrogen, and zinc atoms of AlaNDI-Zn SBCPs are colored gray, white, red, blue, and thin-purple, respectively

in the anti-conformation with respect to the NDI core. All AlaNDI ligands are connected with $Zn(DMF)_2$ units to form a 1D zigzag-chain SBCP. NDIs have strong π-planar stacking ability in the solid state due to their planar aromatic nature[32]. Thus, the zigzag chains are cross-linked to form a 3D framework through strong intermolecular π–π interactions. Each BCP has two centroid-to-centroid distances of NDI planes because of the slightly tilted NDI planes. The centroid-to-centroid distances of NDI planes are 3.58 and 3.72 Å for (*R*)-AlaNDI-Zn, 3.59 and 3.70 Å for (*S*)-AlaNDI-Zn, and 3.61 and 3.63 Å for (*Rac*)-AlaNDI-Zn, respectively. The order of stability in the interaction of two π systems is π-deficient···π-deficient > π-deficient···π-rich > π-rich···π-rich[33]. Therefore, this π-deficient functionality of NDI groups can be used to implement strong, directional π–π stacking

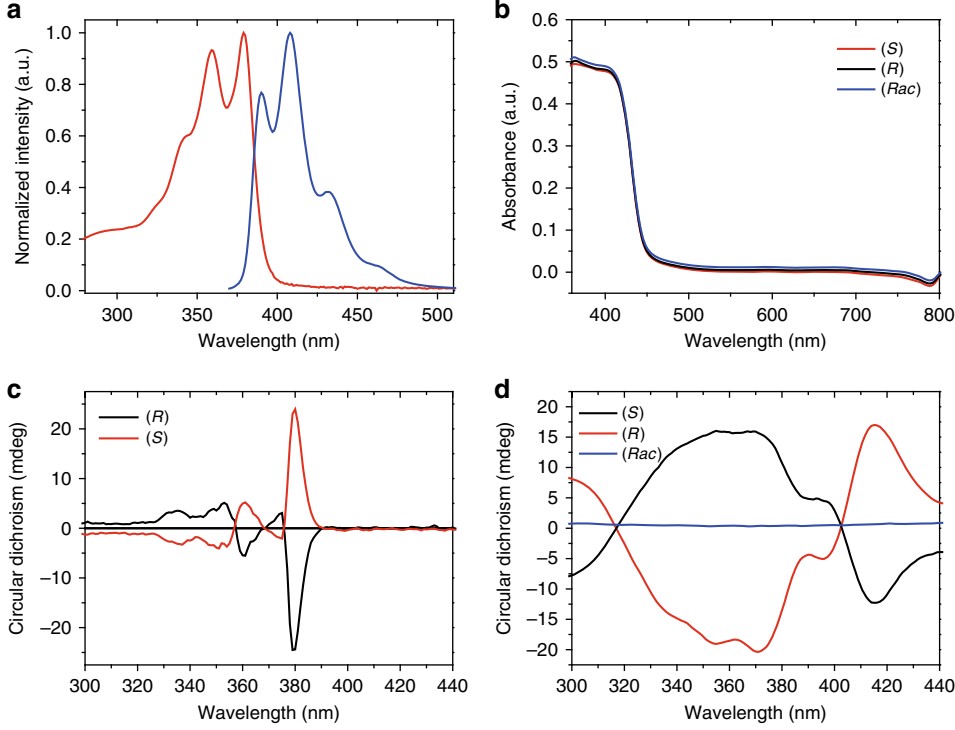

**Fig. 2** UV–Vis spectra, PL spectra, and CD spectra of H$_2$AlaNDI ligands and AlaNDI-Zn SBCPs. **a** Absorption and emission spectra of H$_2$AlaNDI in dichloromethane (1 × 10$^{-4}$ M). **b** UV–Vis spectra of homochiral and heterochiral AlaNDI-Zn SBCPs on a quartz plate. **c**, **d** CD spectra of **c** H$_2$AlaNDI solutions in dichloromethane (1 × 10$^{-4}$ M) and **d** homochiral and heterochiral AlaNDI-Zn SBCPs in ethanol medium

interactions for crystal engineering. These features are commonly observed from all crystals and the homochiral and heterochiral SBCPs show structural similarities.

For a better understanding of the formation of chiral self-discriminated heterochiral crystals when mixing the two enantiomeric ligands, the density functional theory (DFT) calculation of the formation energy ($\Delta E_f$) was carried out using the following equation:

$$\Delta E_f = E_{(Rac)} - \left( E_{(R)} + E_{(S)} \right)/2, \qquad (1)$$

where $E_{(Rac)}$, $E_{(R)}$, and $E_{(S)}$ represent the total energy of unit cell for (Rac)-AlaNDI-Zn, (R)-AlaNDI-Zn, and (S)-AlaNDI-Zn SBCP, respectively (Fig. 1d). The $\Delta E_f$ was calculated to be –1.18 kcal mol$^{-1}$, indicating the formation of heterochiral AlaNDI-Zn is thermodynamically preferable than its homochiral counterpart in excellent accordance with the experimental observations.

**Physical properties of AlaNDI-Zn SBCPs.** The absorption and emission spectra of H$_2$AlaNDI in DMF showed characteristics of monomeric $N,N'$-dialkyl-substituted NDI with well-defined absorption at 360 and 380 nm and mirror-image emission with an approximately 10 nm Stokes shift (Fig. 2a). The absorption peaks of the homochiral and heterochiral AlaNDI-Zn SBCPs showed a strong absorption band at ~400 nm, which can be assigned to the metal-ligand charge transfer transition based on DFT calculations because the highest occupied molecular orbital (HOMO) consists of Zn d$_\pi$ orbitals and the lowest unoccupied molecular orbital (LUMO) consists of ligand π* orbitals (Supplementary Fig. 2). The bandgap ($E_G$) of (R)-AlaNDI-Zn, (S)-AlaNDI-Zn, and (Rac)-AlaNDI-Zn exhibited no significant differences (i.e., 1.58, 1.53, and 1.60 eV, respectively). The strong absorption band centered on 360 nm corresponds to the n–π* and π–π* transitions of aromatic carboxylate ligands (Fig. 2b)[30].

Circular dichroism (CD) is the difference between the absorption of left and right circularly polarized lights[34]. Bisignate Cotton effects were observed for both ligands in dichloromethane (DCM) (1.0 × 10$^{-4}$ M) (Fig. 2c). For example, the CD spectrum of (R)-H$_2$AlaNDI showed a negative peak at 380 nm, whereas that of (S)-H$_2$AlaNDI showed a positive peak at the same wavelength. When the metal ions were coordinated with chromophoric organic ligands, the CD signals could be induced or amplified by combinations of intense chromophoric ligands with suitable metal centers[35]. Figure 2d shows the CD spectra of homochiral and heterochiral AlaNDI-Zn compounds. (S)-AlaNDI-Zn and (R)-AlaNDI-Zn showed mirror-image spectra with opposite-directional peaks, while (Rac)-AlaNDI-Zn showed no discernible CD spectrum. For (S)-AlaNDI-Zn, well-resolved bisignate CD signals (a positive peak at 395 nm and a negative peak at 415 nm) were observed, which may be associated with the exciton-coupled circular dichroism induced by coupling of the excited states of at least two proximal and asymmetrically oriented π–π* chromophores[36]. Thermogravimetric analysis of the homochiral and heterochiral BCPs was also performed under N$_2$ flow at a temperature of 100–800 °C (Supplementary Fig. 3). This result was similar to those of other NDI-based metal-organic framework (MOF)[37]. The weight loss in the range of 100–360 °C originated from the loss of coordinated DMF molecules, and the decomposition of the organic ligand occurred at 400–600 °C.

**Photochromic and photoswitching properties of AlaNDI-Zn SBCPs.** Interestingly, heterochiral AlaNDI-Zn SBCPs showed a photochromic transformation from yellow to dark brown upon UV light irradiation ($\lambda = 365$ nm) for 1 h (Fig. 3a). Power X-ray diffraction (PXRD) (Fig. 3b) and infrared spectroscopy analyses (Supplementary Fig. 4) revealed that the crystal structures were similar before and after irradiation, but their UV–Vis spectra differed. These phenomena indicate that the photo-responsive behaviors may originate from the electron transfer-related

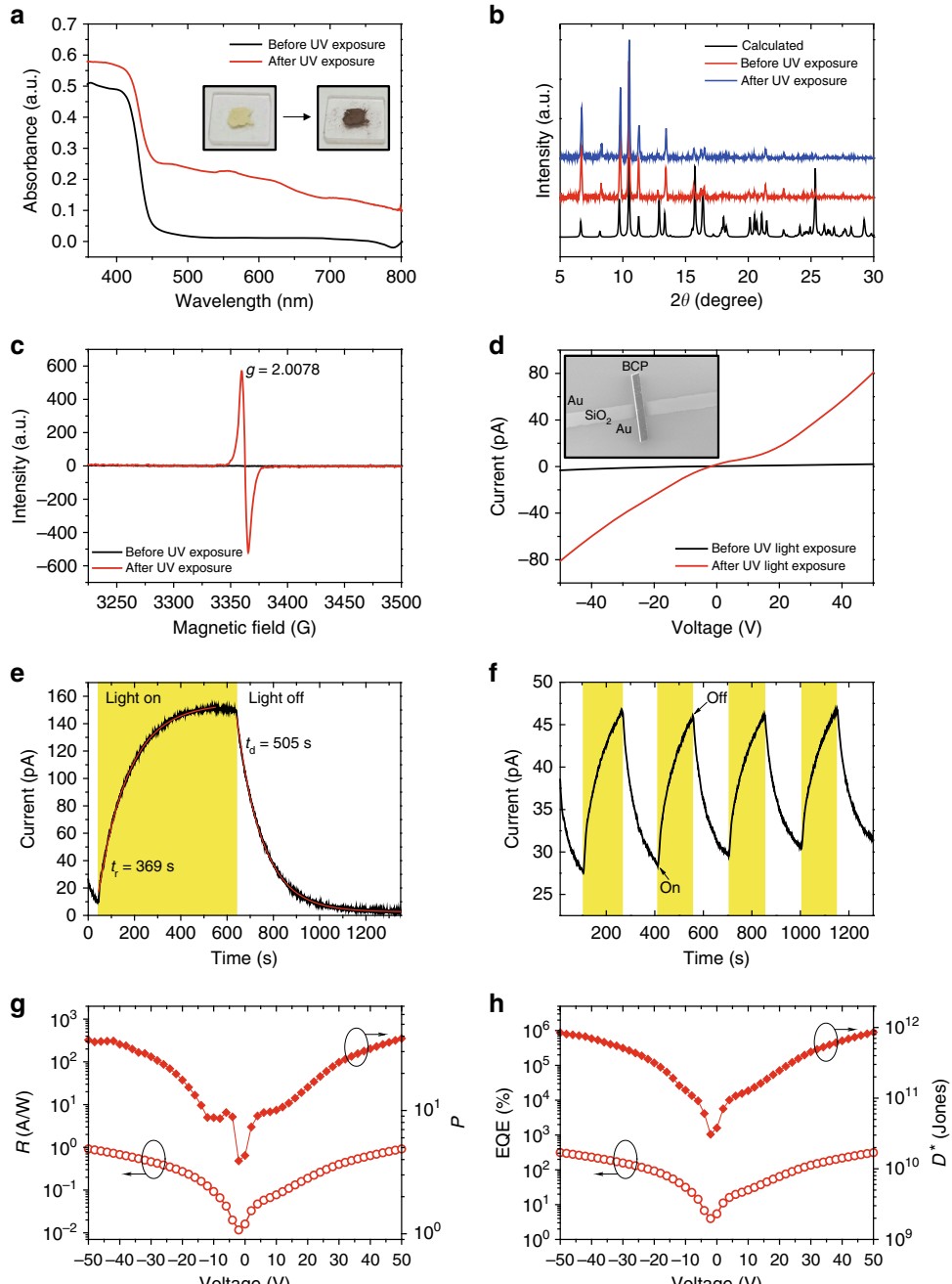

**Fig. 3** Responses of heterochiral (*Rac*)-AlaNDI-Zn SBCPs upon exposure to UV light. **a** UV–Vis spectra of (*Rac*)-AlaNDI-Zn upon exposure to UV light ($\lambda = 365$ nm, 150 μW cm$^{-2}$). Insets show photographs of (*Rac*)-AlaNDI-Zn before and after exposure to UV light for 1 h. **b** Calculated and experimental data of PXRD results for (*Rac*)-AlaNDI-Zn upon exposure to UV light. **c** ESR results of heterochiral (*Rac*)-AlaNDI-Zn SBCPs before and after exposure to UV light for 1 h. **d** *I*–*V* curves of heterochiral (*Rac*)-AlaNDI-Zn SBCPs under UV light illumination. **e**, **f** Time-dependent conductivity changes in heterochiral (*Rac*)-AlaNDI-Zn SBCPs under UV light for **e** rising and decaying time estimations and **f** reversible photoswitching results. **g**, **h** Photoresponsivity (*R*), photocurrent/darkcurrent ratio (*P*), external quantum efficiency (EQE), and detectivity (*D**) of heterochiral (*Rac*)-AlaNDI-Zn SBCPs upon exposure to UV light

chemical process inside the structure rather than the structural transformation[27]. In general, NDI is a redox-active unit that can generate radicals upon light irradiation[32,38]. Therefore, the photochromic process may also arise from the generation of photo-induced radicals in organic ligands[27]. These radical species could be confirmed by electron spin resonance (ESR) spectra (Fig. 3c and Supplementary Fig. 5). No peak was observed before exposure of SBCPs to UV light. With time increased, the intensity of signals in ESR was gradually enhanced within 1 h, indicating

more radicals were formed accordingly (Supplementary Fig. 6). After exposure to UV light for 1 h, SBCPs showed single-peak radical signals with *g* values of 2.0084, 2.0082, and 2.0078 for (*S*)-AlaNDI-Zn, (*R*)-AlaNDI-Zn, and (*Rac*)-AlaNDI-Zn, respectively.

The ability to modulate the properties of CPs on demand by external light-stimuli extends the range of applications[28]. To investigate further the potential of homochiral and heterochiral AlaNDI-Zn SBCPs in optoelectronic applications, their photocurrent changes upon on-switching and off-switching of light

were monitored (Fig. 3d–f). NDI-based materials have been introduced as photoactive materials in phototransistors[39]. However, to the best of our knowledge, there is rare report on photoresponsivity using NDI-based SBCPs. To explore the photoconducting properties of NDI-based SBCP materials, SBCP crystals were drop-casted onto Cr/Au electrodes patterned by photolithography on 300 nm SiO$_2$/Si substrates and their $I$–$V$ characteristics were investigated under ambient conditions (see details in Supplementary Methods). Under illumination of UV light ($\lambda = 365$ nm, 150 µW cm$^{-2}$), SBCPs showed enhanced current in the $I$–$V$ curves. Therefore, this mechanism may be associated with photo-induced electron transfer[32]. Upon light irradiation, additional charge carriers are generated with the formation of radicals that increases the current of SBCPs. When the light is turned off, the radicals return to their original state, resulting in decreased conductivity of SBCPs. The rise/decay in photoresponses of the devices was fitted using exponential rise and decay equations, as follows:

$$I_{light} = I_{dark} + Ae^{t/\tau_{r1}} + Be^{t/\tau_{r2}} \qquad (2)$$

$$I_{light} = I_{dark} + Ae^{-t/\tau_{d1}} + Be^{-t/\tau_{d2}}, \qquad (3)$$

where $t$ is the light switching time, $A$ and $B$ are scaling constants, and $\tau_1$ and $\tau_2$ are time constants for fast and slow rising/decaying rates, respectively. Under UV light illumination (150 µW cm$^{-2}$), heterochiral SBCPs showed very long rising and decaying times of 369 and 505 s, respectively. These changes in photocurrent were reversible, which supports the photoswitchable conductivity of the SBCPs. To further explore the photoresponsive properties, photoresponsivity ($R$), photocurrent/darkcurrent ratio ($P$), external quantum efficiency (EQE), and detectivity ($D^*$) were calculated (see details in Supplementary Information). Heterochiral SBCPs showed quite moderate photoresponsive properties under UV light illumination (Fig. 3g–h). They yielded maximum $R$ and $P$ values of 920 mA W$^{-1}$ and 37 under UV light (150 µW cm$^{-2}$) illumination at an applied bias of 50 V. $D^*$ is an important figure of merit for photodetectors which usually describes the smallest detectable signal, allowing comparison between photodetector devices with different configurations and area. Heterochiral SBCPs exhibited EQE and $D^*$ values of 312% and 8.5 × 10$^{11}$ Jones at an applied bias of 50 V. Unlike the common strategies for photoswitchable CPs that use reversible structural or geometrical changes of the ligand upon light irradiation[28], the developed AlaNDI-Zn SBCPs include photochromic and redox ligand NDI, which can be reversibly converted between a radical state and a neutral state by turning the light on and off, respectively. The resulting high $R$ and $D^*$ values clearly demonstrate the high possibility of the developed CPs for use in optoelectronic applications.

**Application of AlaNDI-Zn SBCPs in photoluminescent sensing**. The metal-organic materials incorporated into NDI ligands are promising for use in sensors[15,40]. For example, NDI-based Ru and Co complexes were used as an infrared probe for monitoring *ds*-DNA[41,42]. Recently, an NDI-based porous Mg-MOF was shown to exhibit selective sensing of amines, which are hazardous to the environment, in a solid state by PL quenching experiments[30].

CPs used for PL sensing generally interact with external analytes in three ways: (1) analyte-luminescent metal ion interaction, (2) analyte-luminescent organic linker interaction, and (3) analyte-trapped luminescent material interaction[30]. Among these, the most efficient synthesis approach is to use the analyte-organic linker interaction since each analyte produces

a distinct signal upon interaction with a luminescent organic linker[30]. Since the AlaNDI-Zn SBCPs are electron-deficient in nature due to the presence of the NDI chromophore, we explored their ability to detect electron-rich hydrazine, which is harmful to the environment. (*Rac*)-AlaNDI-Zn was chosen as a platform for PL sensing. To explore the ability of (*Rac*)-AlaNDI-Zn to sense trace quantities of hydrazine, fluorescence-quenching experiments were performed using analytes with a series of different concentrations of hydrazine in ethanol solution. PXRD showed that (*Rac*)-AlaNDI-Zn can tolerate a concentration of hydrazine in ethanol up to 0.1 M (Supplementary Fig. 7), which was also confirmed by UV–Vis–NIR spectra (Supplementary Fig. 8). The high sensitivity of (*Rac*)-AlaNDI-Zn toward lower concentration of toxic hydrazine was obtained in Fig. 4a and b. The detection limit of (*Rac*)-AlaNDI-Zn for hydrazine in PL sensing was estimated by the following equation:

$$(\text{Detection limit}) = (3 \times S_b)/\text{SL}, \qquad (4)$$

where $S_b$ is the standard deviation value of the measured signal of blank samples (the intensity of (*Rac*)-AlaNDI-Zn in ethanol medium (0.06 mg mL$^{-1}$)), and SL means a slope of linearly fitted sensing results. The estimated detection limit was 3.2 × 10$^{-6}$ M with a correlation coefficient $R^2 = 0.99382$, demonstrating our (*Rac*)-AlaNDI-Zn could detect a trace amount of hydrazine by fluorescence quenching, which was comparable to that of the previous reported PL sensing of hydrazine by NDI-MOF (Fig. 4c)[30]. The fluorescence quenching can be attributed to the donor–acceptor electron transfer between hydrazine and (*Rac*)-AlaNDI-Zn. Based on the DFT calculations, the HOMO and LUMO energy levels of (*Rac*)-AlaNDI-Zn surface (Fig. 4d) indicated that electron transfer from hydrazine to (*Rac*)-AlaNDI-Zn surface was energetically favorable upon light illumination for PL sensing because the HOMO energy (−5.66 eV) of (*Rac*)-AlaNDI-Zn surface is lower than that of hydrazine (−5.47 eV). (*S*)-AlaNDI-Zn with the similar HOMO level showed the similar results to (*Rac*)-AlaNDI-Zn (Supplementary Fig. 9). In addition, the charge transfer amounts from hydrazine to (*Rac*)-AlaNDI-Zn surface tended to linearly increase with increasing surface coverage (Supplementary Fig. 10a), which was similar to experimental results (Fig. 4c). Intriguingly, both the binding energy ($\Delta E_{bind}$) and differential binding energy ($\Delta \Delta E_{bind}$) exhibited a general increasing trend in a stable direction (Supplementary Fig. 10b). This result implied an increased stability by the formation of hydrogen bonds (HB) between adjacent hydrazine molecules (Supplementary Fig. 10c). As a result, the coverage-dependent HB formation at the surface was expected to cause the sensible level of electron transfer.

To our knowledge, the application of homochiral CPs/MOFs have rarely been investigated in enantiodifferentiating fluorescence sensing of chiral analytes[43,44]. To explore the possibility of selectively sensing of homochiral SBCP toward chiral species, we investigated the chiral sensing of homochiral AlaNDI-Zn using naproxen, a nonsteroidal anti-inflammatory drug as the target analyte. The results of the fluorescence quenching titration of AlaNDI-Zn with (*R*)-naproxen or (*S*)-naproxen are shown in Fig. 4e–h. After adding (*R*)-naproxen (10$^{-3}$ M) into (*S*)-AlaNDI-Zn dispersed in ethanol, obvious fluorescence quenching was observed (Fig. 4e). To investigate enantiodifferentiating fluorescence sensing of chiral naproxen, the relative signal intensity ratio ($I/I_0$) of homochiral AlaNDI-Zn upon exposure to different enantiomeric analytes (10$^{-4}$ M) was plotted in Fig. 4f. Interestingly, (*S*)-AlaNDI-Zn exhibited selectively larger fluorescence quenching ability toward (*R*)-naproxen, while (*R*)-AlaNDI-Zn showed the opposite result. Their enantioselective fluorescence

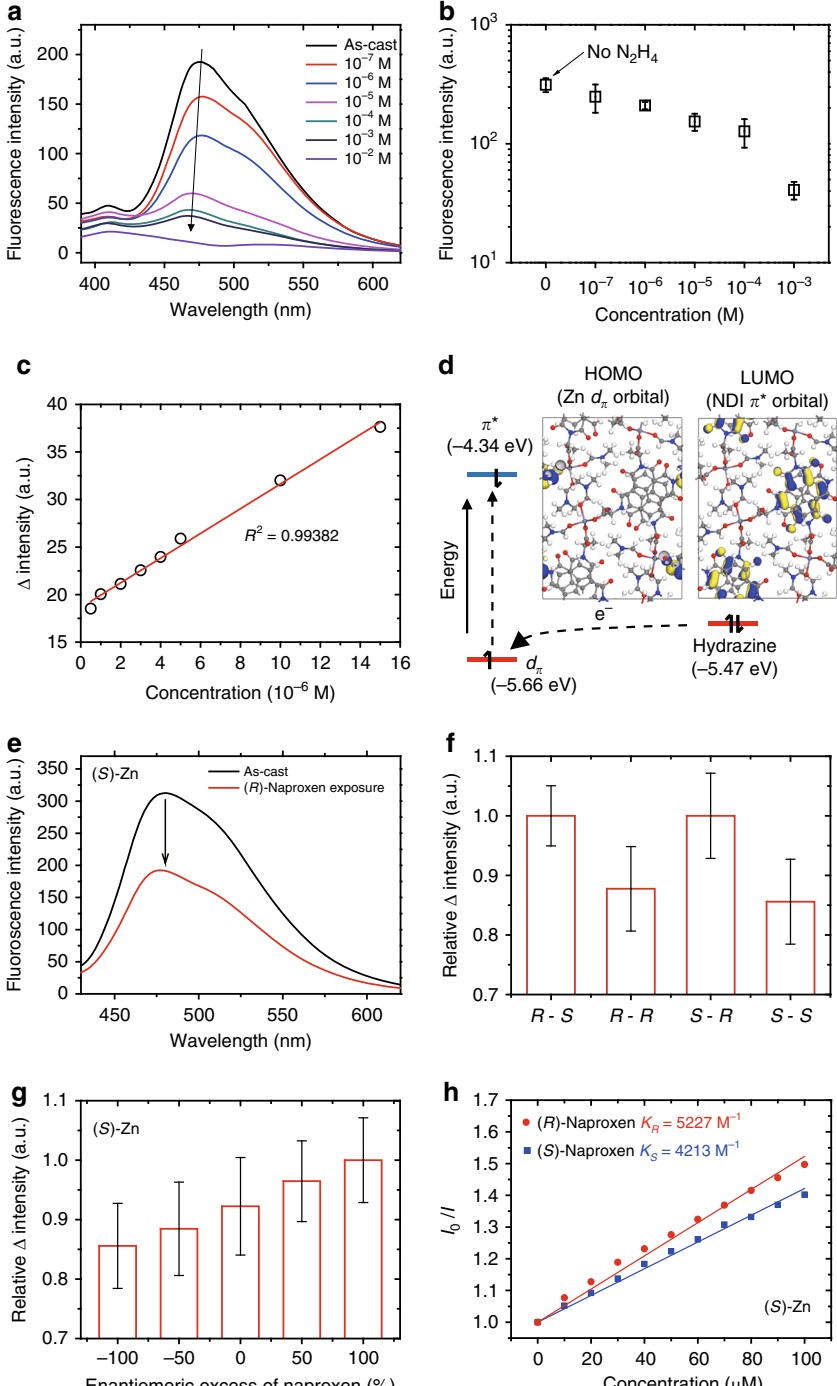

**Fig. 4** Photoluminescent sensing using AlaNDI-Zn SBCPs. **a** Quenching of the fluorescence intensities of (*Rac*)-AlaNDI-Zn dispersed in ethanol medium (0.06 mg mL$^{-1}$) under different concentrations of hydrazine. **b** Fluorescence intensities of (*Rac*)-AlaNDI-Zn dispersed in ethanol medium under different concentrations of hydrazine at $\lambda_{max} = 475$ nm (an excitation wavelength of 360 nm was chosen). **c** Linear fitting of intensity change vs. the concentration of hydrazine at $\lambda_{max} = 475$ nm. **d** Photoluminescent sensing mechanism based on HOMO/LUMO formalism of the heterochiral AlaNDI-Zn SBCP surface and hydrazine. The carbon, hydrogen, oxygen, nitrogen, and zinc atoms of AlaNDI-Zn SBCPs are colored gray, white, red, blue, and thin-purple, respectively. **e** Quenching of the fluorescence intensity of (*S*)-AlaNDI-Zn under (*R*)-naproxen (10$^{-3}$ M) in ethanol medium. **f** Relative fluorescence intensities of AlaNDI-Zn dispersed in ethanol medium under different chiralities of naproxen (10$^{-4}$ M) at $\lambda_{max} = 475$ nm, e.g., *R-S* means ((*R*)-AlaNDI-Zn-(*S*)-naproxen). **g** Relative fluorescence intensities of AlaNDI-Zn dispersed in ethanol medium under different enantiomeric excesses of naproxen (10$^{-4}$ M) at $\lambda_{max} = 475$ nm. **h** Linear Stern–Volmer plots for the enantiodifferentiating fluorescence sensing of (*S*)-AlaNDI-Zn dispersed in ethanol medium for (*S*)-naproxen and (*R*)-naproxen. The average PL intensities were taken from the 50 data points collected in different five samples for all PL sensing experiments

quenching was further investigated for the potential application to quantify the enantiomeric excess of the chiral analytes. The relative fluorescence intensity ratio of AlaNDI-Zn was tested under different enantiomeric excess of naproxen exposure ($10^{-4}$ M) in ethanol medium at $\lambda_{max} = 475$ nm (Fig. 4g). The relative quenching of $I/I_0$ was linearly correlated with the enantiomeric composition of naproxen. Therefore, the enantiomeric excess of naproxen could be easily determined by a simple fluorescence quenching detection.

The fluorescence quenching data for both (R)-naproxen and (S)-naproxen to (S)-AlaNDI-Zn SBCPs can be readily subjected to linear regression analysis using Stern–Volmer equation:[45]

$$I_0/I = 1 + K_{sv}[Q]. \tag{5}$$

$I_0$ is the inherent fluorescence intensity of (S)-AlaNDI-Zn, $I$ is the fluorescence intensity in the presence of the quencher (R)-naproxen or (S)-naproxen, $K_{SV}$ is the Stern–Volmer constant. Based on a linear Stern–Volmer plot, $K^R_{SV}$ and $K^S_{SV}$ were estimated to be 5227 and 4213 $M^{-1}$ for (R)-naproxen and (S)-naproxen, respectively (Fig. 4h). Accordingly, the enantiodiscrimination of $\Delta K$ and enantioselectivity factor $\alpha$ were also determined to be 1014 $M^{-1}$ and 1.24, respectively, from the following equations:

$$\Delta K = \left( K^R_{SV} - K^S_{SV} \right) \tag{6}$$

$$\alpha = K^R_{SV}/K^S_{SV}. \tag{7}$$

To further verify the origin of enantioselectivity of homochiral SBCPs, the binding interactions of chiral naproxen molecules were theoretically compared (Supplementary Figs. 11, 12). As a result, the (R)-naproxen exhibited a stronger binding interaction with (S)-AlaNDI-Zn than its (R)-counterpart with increasing surface coverage, whereas the (S)-naproxen showed the opposite trend, as confirmed by the experimental results (Fig. 4f, h). Therefore, the binding energy differences of chiral analytes at the surface are predicted to be the origin of enantioselective adsorption.

**Application of AlaNDI-Zn SBCPs in chemiresistive sensing**. The use of CPs as chemical sensors has attracted great attention in recent years due to their high surface area and chemical tunability[20,21]. However, one important challenge is to overcome the difficulty in the efficient signal transduction, which is related to the fact that the majority of CPs are insulators. To the best of our knowledge, few reports are available for applications in chemical sensors using the electrical responses of CPs. Recently, a conductive 2D MOF was used for chemiresistive sensing[22,23]. An NDI-based material has been used to detect alcohols, such as ethanol[46]; however, there is no research on chemiresistive sensing of electron-rich alcohols and amines using insulative NDI-based BCPs. In our study, the (Rac)-AlaNDI-Zn SBCPs were drop-casted onto gold electrodes-patterned SiO$_2$/Si substrates and used as chemiresistive sensors to selectively detect electron-rich VOCs. I–V curves of (Rac)-AlaNDI-Zn chemiresistive sensors upon exposure to various saturated gaseous VOCs are shown in Fig. 5a. The conductivity was increased in the presence of electron-rich analytes, such as methanol, ethanol, and aniline, while the conductivity was kept almost constant for other common solvents such as DCM and n-hexane (Fig. 5b). The sensitivities toward analytes were compared by measuring the enhanced current with regard to the concentration of the exposed VOC vapors. In the quantitative comparison, (Rac)-AlaNDI-Zn showed similar

sensitivity under ethanol and methanol saturated vapors. Interestingly, the sensitivity for aniline was much higher than that for alcohol (>2 orders of magnitude), due to the stronger electron donating ability of aniline than alcohols. To theoretically elucidate the chemiresistive sensing mechanism, we investigated the electronic density of states (DOS) before and after adsorption of aniline at the (Rac)-AlaNDI-Zn surface by DFT calculation (Fig. 5c–e). Due to the adsorption of electron-rich aniline, the peak intensity at the HOMO level corresponding to the π orbital of aniline was decreased, implying the electron donating behavior from aniline to the (Rac)-AlaNDI-Zn surface. Concurrently, the LUMO level corresponding to the π* orbital of NDI ligands was downshifted, thereby resulting in the reduction of energy gap, which was demonstrated to be the origin of enhanced conductivity.

In order to further clarify the strong detection ability of the amine, we performed real-time sensing measurements using aniline as the target analyte. For this experiment, the stream of aniline vapor with different concentration was continuously passed over a device at a constant rate of 7.0 L min$^{-1}$ and the current variation was monitored (Fig. 5f). (Rac)-AlaNDI-Zn SBCPs showed a relatively constant current baseline before they were exposed to the aniline vapor. However, the current was sharply increased in a couple of seconds when the device was exposed to the aniline vapor (Fig. 5f, inset). As the concentration of aniline increased, the current was also enhanced accordingly. The detection limit of aniline vapor was approximately 16 ppm (according to the calculation using Eq. (4) and Supplementary Fig. 13), which was comparable to those of conductive CPs reported previously for sensing ammonia or amines[22,23]. The initial current level was recovered when the stream of aniline vapor was turned off. To further demonstrate the stability of the devices under exposure to the aniline vapor, PXRD analysis was conducted (Fig. 5g). The similar signals before and after exposure to the saturated aniline vapors suggest that the crystallinity of (Rac)-AlaNDI-Zn was not affected by the aniline analyte. Notably, the gaseous aniline at low concentration could be detected under ambient conditions, which suggests great potential of the developed insulative CPs for use in practical applications.

**Discussions**

In conclusion, we synthesized homochiral and heterochiral AlaNDI-Zn SBCPs from pure and mixed alanine-based NDI enantiomeric ligands, respectively. The selective formation of heterochiral AlaNDI-Zn was due to their thermodynamic preference with lower formation energy than their homochiral counterpart. These SBCPs were successfully used for PL sensing of toxic hydrazine with a detection limit as low as $3.2 \times 10^{-6}$ M. The sensing mechanism can be attributed to the donor–acceptor electron transfer between hydrazine and the SBCP. Judging from DFT calculation, the HOMO energy of heterochiral SBCP surface (−5.66 eV) is lower than that of hydrazine (−5.47 eV), thus such electron transfer can occur under light illumination. Interestingly, the photoluminescent homochiral SBCPs has been successfully developed to explore the chiral acid naproxen by fluorescence quenching with high sensitivity and selectivity owing to the different binding energies of chiral analytes at the SBCP surface. Furthermore, we demonstrated that the insulator SBCPs can be used to selectively detect common VOCs, such as methanol, ethanol, and aniline. This sensing capability originates from the strong electron-donating abilities of alcohols or aniline when they interact with the electron-deficient moiety of NDI ligand of the SBCP, which leads to the reduction of the energy gap for the chemiresistive sensing. Among the tested VOCs, aniline showed

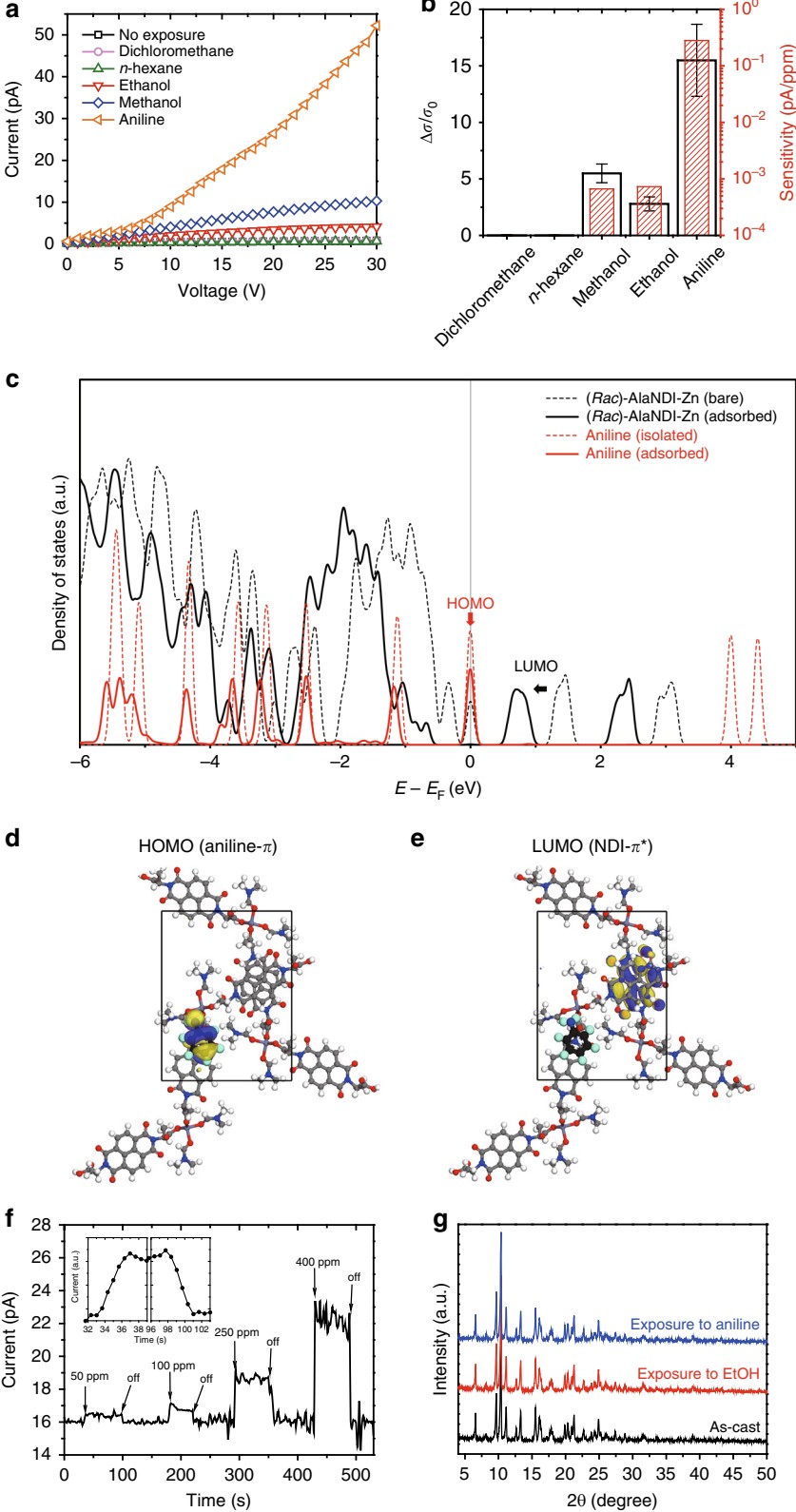

the highest sensitivity with a detection limit of approximately 16 ppm, which is comparable to the reported chemiresistive detection limit of ammonia or amines using the conductive CPs[22,23]. These results suggest that the developed CPs with low conductivity have great potential for use in chemical sensors. Besides,

the photochromic SBCPs exhibited a color change from yellow to dark brown upon UV light irradiation for 1 h because of the electron transfer-related chemical process inside the structure as well as the generation of photo-induced radicals in the NDI ligands, according to the PXRD, UV–Vis spectroscopy, and

**Fig. 5** Chemiresistive sensing using heterochiral AlaNDI-Zn SBCPs and their electronic structure analysis of the heterochiral AlaNDI-Zn SBCP surface by aniline adsorption. **a** I–V curves of heterochiral SBCPs upon exposure to the vapors of various analytes. **b** Relative responses and sensitivities of the (*Rac*)-AlaNDI-Zn device upon exposure to various vapors. The average conductivity (σ) changes were taken from the 30 sensing experiments measured for different 5 devices. **c** Electronic density of states (DOS) of aniline (red line) and the (*Rac*)-AlaNDI-Zn surface (black line) before (dotted line) and after (solid line) aniline adsorption. The red and black arrows represent the shifts in HOMO and LUMO levels, respectively. **d** Electronic structure of HOMO level consisting of the π orbital of aniline. **e** Electronic structure of LUMO level consisting of the π* orbital of the NDI ligand. The carbon, hydrogen, oxygen, nitrogen, and zinc atoms of (*Rac*)-AlaNDI-Zn are colored gray, white, red, blue, and thin-purple, respectively. The carbon, hydrogen, and nitrogen atoms of aniline are represented as large spheres colored black, cyan, and blue, respectively. **f** Real-time responses of the (*Rac*)-AlaNDI-Zn device upon exposure to aniline vapor with different concentrations at a constant voltage of 100 V. Inset shows the sensing speed under exposure to aniline gas at 50 ppm in short time ranges. **g** PXRD results of heterochiral AlaNDI-Zn crystals after exposure to saturated aniline and ethanol gas for 30 min

ESR spectroscopy analyses. Furthermore, we observed the photoswitching behaviors of the SBCPs due to the generation of photo-induced radicals in the redox-active NDI ligand, which differs from other photoswitchable CPs showing structural or geometrical changes of the ligand upon light irradiation. This result makes them more suitable for use in optoelectronic devices. The SBCPs under UV light illumination (150 μW cm$^{-2}$) yielded well-resolved photophysical responses with the EQE, $D^*$, maximum $R$ and $P$-values of 312% and $8.5 \times 10^{11}$ Jones, 920 mA W$^{-1}$, and 37, respectively, at an applied bias of 50 V. In addition, the developed SBCPs can selectively exhibit chiral self-sorting phenomena by rationally controlling the synthetic conditions. Considering the research on the sensor applications of chiral CPs is still in its infancy, the multifunctional chiral SBCPs developed herein will expedite the practical applications of CPs in various sensors.

## Methods

**Synthesis of homochiral H$_2$AlaNDI ligands.** Homochiral ligands were synthesized from NTCDA (1.34 g; 5 mmol) and L-alanine or D-alanine (0.89 g; 10 mmol), which were refluxed for 12 h in pyridine (600 mL). When the volume of the mixture was reduced to 10 mL, the aq HCl (300 mL water and 100 mL conc. HCl) was added into the mixture. The solid precipitate was separated by filtration[31].

**Synthesis of (*R*)-AlaNDI-Zn and (*S*)-AlaNDI-Zn SBCPs.** ZnI$_2$ (0.1 mmol) and chiral H$_2$AlaNDI (0.1 mmol) were mixed in 3 mL DMF and sonicated for 30 min at room temperature. This solution was sealed in a stainless-steel tube with a Teflon liner and then heated at 120 °C for 72 h. The crude product was filtered and washed with DMF to give the final single crystals. The yield of the colorless single crystal of (*R*)-AlaNDI-Zn was ca. 27%. Anal. Calcd. for C$_{26}$H$_{26}$ZnN$_4$O$_{10}$ (%): C 50.38, H 4.23, N 9.04; found (%): C 50.07, H 4.11, N 8.72. The yield of the colorless single crystal of (*S*)-AlaNDI-Zn was ca. 29%. Anal. Calcd. for C$_{26}$H$_{26}$ZnN$_4$O$_{10}$ (%): C 50.38, H 4.23, N 9.04; found (%): C 50.13, H 4.10, N 8.70.

**Synthesis of (*Rac*)-AlaNDI-Zn SBCPs.** ZnI$_2$ (0.1 mmol), (*R*)-H$_2$AlaNDI (0.05 mmol), and (*S*)-H$_2$AlaNDI (0.05 mmol) were homogenized in 3 mL DMF and sonicated for 30 min at room temperature. This solution was sealed in a stainless-steel tube with a Teflon liner and then heated at 120 °C for 72 h. The crude product was filtered and washed with DMF to give the final single crystals. The yield of the colorless single crystal of (*Rac*)-AlaNDI-Zn was ca. 21%. Anal. Calcd. for C$_{26}$H$_{26}$ZnN$_4$O$_{10}$ (%): C 50.38, H 4.23, N 9.04; found (%): C 50.05, H 4.22, N 8.93.

**Device fabrication.** Heavily *n*-doped silicon wafers (<0.004 Ω cm) with thermally grown 300-nm-thick SiO$_2$ ($C_i$ = 10 nF cm$^{-2}$) were used as substrates for electronic devices. The wafers were washed with toluene, acetone, and isopropyl alcohol, and dried with nitrogen gas. The Cr/Au electrodes (4/40 nm) were thermally evaporated and patterned using conventional photolithography on the substrate. The electrode patterns had a channel length ($L$) of 10 μm and a channel width ($W$) of 200 μm ($W/L$ = 20). To fabricate the electronic devices, the AlaNDI-Zn crystals dispersed in DMF were drop-casted on the substrate. The substrates were annealed at 60 °C in a vacuum oven to evaporate the residual DMF.

**Electrical measurements.** The current–voltage characteristics of the SBCPs were measured in ambient conditions using a Keithley 4200-SCS parametric analyzer.

**Material analyses.** Elemental analyses were performed on an Elementar vario MICRO cube at a Technical Support Center in POSTECH. The absorption spectra were measured on a Cary 5000 UV–Vis–NIR and a Cary 6000i UV–Vis–NIR spectrophotometer for ligands and SBCPs, respectively. PL spectra were recorded

on an FP-6500 spectrofluorometer (JASCO). The CD results were obtained using a J-815 Spectropolarimeter (JASCO). ESR spectra were measured on a Bruker Biospin A200 spectrometer. PXRD results were obtained using a Bruker Advance D8 instrument.

**Crystal structure analyses.** The single-crystal XRD data was collected on a Bruker APEX II QUAZAR instrument. All the structures were solved by direct methods (SHELXS-97/SHELXS-2014) and refined by full-matrix least squares calculations on $F^2$ (SHELXL-2014) using the SHELX-TL program package.

**Crystallographic data for (*R*)-AlaNDI-Zn single crystal.** C$_{52}$H$_{52}$N$_8$O$_{20}$Zn$_2$ $M_r$ = 1239.75, crystal dimensions $1.00 \times 0.20 \times 0.10$ mm$^3$, monoclinic, space group $P2_1$, $a$ = 7.293(3), $b$ = 21.603(8) Å, $c$ = 16.845(6) Å, $\beta$ = 91.665(12)°, $V$ = 2652.8(17) Å$^3$, $Z$ = 2, $\rho_{calcd}$ = 1.552 g cm$^{-3}$, $\mu$ = 0.992 mm$^{-1}$, $\lambda$ = 0.71073 Å (Mo Kα), $T$ = 100(2) K, 5512 reflections out of 7575 with $I > 2\sigma(I)$, 710 parameters, 823 restraints, 1.21 < $\theta$ < 25.45°, final $R$ factors R1 = 0.0576 and wR2 = 0.1021, GOF = 1.036. Flack parameter: 0.03(3). CCDC: 1529614.

**Crystallographic data for (*S*)-AlaNDI-Zn single crystal.** C$_{52}$H$_{52}$N$_8$O$_{20}$Zn$_2$ $M_r$ = 1239.75, crystal dimensions $0.90 \times 0.30 \times 0.20$ mm$^3$, monoclinic, space group $P2_1$, $a$ = 7.278(1), $b$ = 21.582(2) Å, $c$ = 16.822(1) Å, $\beta$ = 91.361(1)°, $V$ = 2641.6(4) Å$^3$, $Z$ = 2, $\rho_{calcd}$ = 1.559 g cm$^{-3}$, $\mu$ = 0.996 mm$^{-1}$, $\lambda$ = 0.71073 Å (Mo Kα), $T$ = 100(2) K, 7828 unique reflections out of 9642 with $I > 2\sigma(I)$, 752 parameters, 781 restraints, 1.21 < $\theta$ < 23.38°, final $R$ factors R1 = 0.0493 and wR2 = 0.0932, GOF = 1.021. Flack parameter: 0.044(19). CCDC: 1529613.

**Crystallographic data for (*Rac*)-AlaNDI-Zn single crystal.** C$_{52}$H$_{52}$N$_8$O$_{20}$Zn$_2$ $M_r$ = 1239.75, crystal dimensions $0.90 \times 0.20 \times 0.20$ mm$^3$, monoclinic, space group $P2_1$, $a$ = 7.240(1), $b$ = 21.628(3) Å, $c$ = 16.859(2) Å, $\beta$ = 91.636(6)°, $V$ = 2638.7(5) Å$^3$, $Z$ = 2, $\rho_{calcd}$ = 1.560 g cm$^{-3}$, $\mu$ = 0.997 mm$^{-1}$, $\lambda$ = 0.71073 Å (Mo Kα), $T$ = 100(2) K, 5092 unique reflections out of 9145 with $I > 2\sigma(I)$, 740 parameters, 825 restraints, 1.21 < $\theta$ < 24.90°, final $R$ factors R1 = 0.0708 and wR2 = 0.1415, GOF = 1.007. Flack parameter: 0.31(5). CCDC: 1529612.

**Model systems for calculations.** The unit cell structures of homochiral and heterochiral AlaNDI-Zn SBCPs obtained from experimental XRD analysis were optimized by DFT calculation while keeping the experimental lattice parameters. Next, the surface systems were modeled by introducing the vacuum in the (100) direction, where the AlaNDI-Zn fragments were stacked. Note that to fill the dangling sites of Zn atoms, two AlaNDI-Zn ligands were added (i.e., 356 atoms). By Monte Carlo simulation, the initial binding configuration in a fixed loading of analytes (i.e., hydrazine, aniline, and chiral naproxen) were found by sequentially adding the analyte molecules on the bare AlaNDI-Zn surfaces. Subsequently, the optimized structures were obtained by full relaxation via the DFT calculation while keeping the crystallographic positions of all atoms in the AlaNDI-Zn surfaces.

**DFT calculations.** All DFT calculations were performed with DMol$^3$ program[47,48] with the generalized gradient approximation with the Perdew–Burke–Ernzerhof functional[49], in which the semi-empirical Grimme scheme[50] for the dispersion correction was included. Spin-polarized calculations were performed with the basis set of DNP 4.4. DFT semi-core pseudopotentials were applied for all model sysems. The Brillouin-zone was sampled by a Monkhorst-Pack[51] as Γ-point for all model systems. The convergence criteria for energy, force, and displacement were set as $1.0 \times 10^{-5}$ Ha, 0.002 Ha Å$^{-1}$, and 0.005 Å, respectively. The atomic charges were obtained from Mulliken population analysis[52]. For the DOS analysis, the smearing width was set to be 0.05 eV.

**Monte Carlo simulations.** Monte Carlo simulations with fixed number of analytes (i.e., hydrazine, aniline, and naproxen) were carried out by Sorption program[53] to find the preferential adsorption sites on AlaNDI-Zn surfaces. For the interaction energy parameters, Universal forcefield[54] was employed with atomic charges obtained from Mulliken population analysis. The systems were equilibrated for 1 ×

$10^6$ MC steps and analyzed for the next $1 \times 10^6$ MC steps. The non-bond interactions, including electrostatic and van der Waals forces, were estimated by Ewald summation and atom-based cut-off (a radius of 12.5 Å) scheme.

## Data availability

The authors declare that the all data supporting the findings of this study are available within this article and Supplementary Information files, and also are available from the authors upon reasonable request. CCDC 1529612, 1529613, and 1529614 contain the crystallographic data for this paper. These data can be obtained free of charge from The Cambridge Crystallographic Data Centre via www.ccdc.cam.ac.uk/getstructures. The data that support the findings of this study are available from the corresponding authors upon reasonable request.

Published online: xx xxx 2018

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

## Acknowledgements

This work was supported by the Samsung Research Funding Center of Samsung Electronics under Project Number SRFC-MA1602-03. Single-crystal XRD experiments at PLS-II were supported in part by MSICT and POSTECH. S.K.K. acknowledges the computational resources from UNIST-HPC.

## Author contributions

X.S. and J.H.O. conceived the idea. J.H.O. directed all the research project. X.S. synthesized the materials. I.S. performed device fabrication and electrical characterizations. X.S., I.S. and J.A. performed instrumental analyses such as UV, PL, and CD measurements. G.Y.J., J.H.L. and S.K.K. carried out the DFT calculation. W.C. and H.O. performed XRD measurement. W.C., B.L., H.O. and M.K. solved the crystal structures. J.Y. K. assisted in ESR measurement. X.S., I.S., S.K.K. and J.H.O. wrote the manuscript, and all authors commented on the manuscript.

## Additional information

**Competing interests:** The authors declare no competing interests.

