## [Peer Review File · Nature Communications]

Reviewers' comments:

Reviewer #1 (Remarks to the Author):

The authors report novel materials designs of homochiral and heterochiral supramolecular biocoordination polymers. The materials are thoroughly characterized in terms of their crystal structures, electronic structures and optical properties. These materials exhibit pronounced photochromic and photoswitchable behavior without altering the crystal structures. This optical property enabled PL sensing of electron-rich amine via photoinduced electron transfer and radical generation. The authors further report chemiresistive sensing of VOCs, despite that the materials were insulating. Interestingly, upon exposure to electron-rich vapors, the insulating materials became conductive. Both the molecular designs and the physical properties are novel, and therefore the work should be considered favorably for publication in Nature Communications if the following comments are addressed.

- 1) How does the chirality contribute to the photoswitchability and the chemiresistivity observed? It appears that both the hetero and homo-chiral molecules exhibited these properties and have been used towards sensing devices. It is unclear what design principles the authors followed to impart the specific properties, and why alanine linker or chirality is necessary at all.
- 2) In the introduction, the authors discussed the challenges of fabricating heterochiral structures. Could the authors elaborate what design considerations were taken to yield the heterochiral SBCP reported in this work?
- 3) The increased current upon exposure to electron-rich vapors is attributed to formation of the charge transfer complex. The formation of charge transfer complex may not directly contribute to increase in conductivity. Furthermore, it will be helpful to provide evidence of the charge transfer complex formation, using UV-Vis-NIR spectroscopy for instance, or through DFT calculations built upon the DFT studies already performed.
- 4) Regarding Figure 4, the authors claim that the detection limit to H₂H₄ is 10⁻⁷ M. Typically, the detection limit is not equal to the lowest concentration tested. Instead, the detection limit can be extrapolated from the linear calibration curve when the signal equals three times the root-mean-square deviation of the noise level.

Reviewer #2 (Remarks to the Author):

The manuscript entitled 'Chiral Self-Discriminated Multifunctional Supramolecular Biocoordination Polymers (SBCPs) and Their Applications in Sensors' by Shang et al. deals with biocoordination polymer of Naphthalenediimide (NDI) with amino acid functionalities. NDI with alanine-termini (D/L) was coordinated with zinc ion that forms homo, heterochiral, and also enantiomeric derivative with mixed R- and S-ligands. Such a property has been well studied in the literature and many of the original references are not cited. The synthesis was carried out following previous literature. They successfully formed single crystal structures of the different chiral ALANDI-Zn derivatives. The multifunctional property of the SBCPs has been highlighted in this manuscript. This work includes measuring number of material properties and completely lacks novelty and direction required for a high impact journal. As the author claimed four main properties – photochromic, photoluminescent, photoconductive, and chemiresistive, but none of them are correlated. It would be better to focus on a specific or related properties and functionalities. All the application aspects are well reported in previous literatures. The homo- vs. heterochirality has not been discussed in depth in the manuscript, and also the author failed to connect the chirality to the functionality and not discussed with reference to known and similar studies from the literature, that would give the reader good perspective of the work.

Authors claimed in page-3, para-2nd: "However, the π - π interaction have not been well characterized." is not acceptable. There are several reports on this topic. Again, in page-4, para-2nd: ".....CPs as next-generation functional materials in electronics and photonics." – the

statement, however, was not justified in their work. They reported a simple photo-current generation which is well known for NDI derivatives, as they have electro-potential similar to chlorophyll. The DFT calculation is also not very informative – for these types of systems already the PET mechanism is well known. If the author can specifically comment on the chirality-induced photocurrent differences and their applications then it will be interesting. Moreover, in the experimental section, they reported the activation time for photo-current is about 1 hour, which is as slow as light-induced experiments are very fast. It seems that material is photo-resistive, that's why they mentioned it as photo-sensitive. I am more curious to know what happened before getting a single peak in EPR. The time-lag has not been elucidated.

The author mentioned different molecular packing in (S/R)-AlaNDI-Zn and (Rac)-AlaNDI-Zn, but the information was not extended to the different properties. The last two sections, PL sensing and chemiresistive – in my opinion, are not required. There is no significant correlation with other parts of the report.

In conclusion, I read the manuscript very carefully and found it lacks novelty and direction. Too many independent topics finally made it unclear and confusing. Therefore, work is not suitable for publication in Nature Communications and suitably revised manuscript may be considered for Scientific Reports.

Reviewer #3 (Remarks to the Author):

Hak Oh et al report on the use of supramolecular coordination polymers based on the coordination of Zn(II) ions with electroactive naphthalene diimide (NPDI) molecules functionalized with chiral alanine residues. This coordination polymers are then used to test different properties like photoconductivity, photoluminescence or chemiresistive sensing.

After reading the paper it is hard to understand why the authors give so much importance to the chirality of the polymers as its influence over the properties highlighted is not explored. In my opinion, the use of chiral analytes for their experiments on sensing –either by quenching of the luminescence or electrical signal transduction– would have been substantially more interested than those presented in the work. In the current state, chirality appears of vague added value rather than reinforcing the interest of the work. Even more so, when chiroptical sensing with similar systems has been recently reported by the authors (*Adv. Mater.* 2017, 29 (21), 1605828). It is hard to understand why the author give this so much importance in the introduction and their discussion if chirality is not studied in any way.

I don't recognise substantial novelty of broad interest and would recommend publication somewhere else.

Photoconductivity of NPDI molecules has been already reported (*Appl. Phys. Lett.* 103, 053301 (2013) & *ACS Appl. Mater. Interfaces*, 2014, 6 (15), pp 12295–12301). As for the sensing based on photoluminescence quenching there are plenty of reports (some are referenced by the authors) and this case seems to be more relevant because the use of a toxic analyte like hydrazine. Still the CP is not porous and it is unclear to me how the sensing operates. There are no pores in the structure to accommodate the guest to induce electron transfer that might be instead limited to the surface. In fact, the authors compare their result to a porous system NDI-MOF (ref 33) and claim that works better but it is difficult to understand why. Also, supplementary Fig 6 shows substantial amorphization of the solid in contact with hydrazine that might be the actual responsible for the observed quenching of the signal (interaction with NPDI molecules in solution upon progressive degradation of the solid). My concerns on the reported chemiresistive sensing are also based on the comparison with previous reports of similar experiments based on porous MOFs by Dinca et al. (ref 25 and 26). Again, it is hard to understand how the guest will interact

with their solid beyond mere charge transfer at a superficial level. The authors do not comment on the stability of their material during the experiments (there is no data that helps evaluating this point) and it is difficult to rule out if the behaviour might be linked to NPDI molecules in solution.

Also important, the properties reported are intimately linked to the use of NPDI connectors assembled into a 3D solid. Still, the authors invoke DFT calculations based on simple clusters rather than modelling the periodic structures. This is key to establish an origin for the observed phenomena but I think VASP methodologies with periodic boundary conditions would be more adequate to establish clear correlations with the electronics structure of the solid rather than the HOMO/LUMO formalism used, which appears more adequate for isolated molecules. Their approach is probably mistreating the effect that π - π stacking between neighbouring NPDI molecules (clearly observed from SCXR data) might have over the electronic structure. This is key to the work as it is expected to dictate both photoconductivity and chemiresistive sensing.

COMMENTS TO AUTHOR:

Reviewer #1

Comments:

The authors report novel materials designs of homochiral and heterochiral supramolecular biocoordination polymers. The materials are thoroughly characterized in terms of their crystal structures, electronic structures and optical properties. These materials exhibit pronounced photochromic and photoswitchable behavior without altering the crystal structures. This optical property enabled PL sensing of electron-rich amine via photoinduced electron transfer and radical generation. The authors further report chemiresistive sensing of VOCs, despite that the materials were insulating. Interestingly, upon exposure to electron-rich vapors, the insulating materials became conductive. Both the molecular designs and the physical properties are novel, and therefore the work should be considered favorably for publication in Nature Communications if the following comments are addressed.

1) How does the chirality contribute to the photoswitchability and the chemiresistivity observed? It appears that both the hetero and homo-chiral molecules exhibited these properties and have been used towards sensing devices. It is unclear what design principles the authors followed to impart the specific properties, and why alanine linker or chirality is necessary at all.

Response: Thank you for your valuable comments. The design principle of this work was inspired by our earlier study (*Adv. Mater.* **2017**, *29*, 1605828). Interestingly, upon mixing the *R*- and *S*-CPDI-Ph enantiomers, we observed the formation of heterochiral NWs, i.e. chiral self-discrimination. Therefore, we presumed that heterochiral coordination polymers may also be formed by mixing the two enantiomeric ligands and the metal source. The alanine linker was used as the best choice for enhancing the crystallinity of BCPs, because various other amino acids tested as linkers could not form crystals that are good enough for XRD measurement. In order to explore the contribution of chirality to photoswitchability and chemiresistivity, we initially tried the circularly polarized light (CPL) detection and chiral chemiresistive sensing using homochiral BCPs, but unfortunately we failed to get any meaningful data regarding the effects of chirality on these properties. This was the reason why we rather focused on the chiral self-discrimination of heterochiral crystals and their use in

multifunctional sensors, since these properties themselves have rarely been reported. According to your valuable comments, we additionally performed chiral photoluminescent sensing of a chiral analyte naproxen, which is a nonsteroidal anti-inflammatory drug of the propionic acid class that relieves pain, fever, swelling, and stiffness, using homochiral BCPs, in order to investigate the effects of chirality on sensing behaviors. Also, we compared the binding interactions of chiral naproxen molecules at the homochiral BCP surfaces by DFT calculations to identify the origin of enantioselectivity. From these additional experiments and theoretical calculations, we obtained very interesting results on the enantioselective chemical detection using photoluminescence quenching phenomena and revised the manuscript as follows:

On page 5

Interestingly, our homochiral SBCP can enantioselectively detect chiral naproxen by fluorescence quenching with high sensitivity because of their different binding strengths. Furthermore, we have used insulating SBCPs for chemiresistive sensing of electron-rich VOCs, such as methanol, ethanol, and aniline with enhanced conductivity due to the reduction of energy gap. The detection limit for electron-donating aniline reaches 16 ppm with the insulative SBCPs, which is comparable to the reported high-performance chemiresistive sensing for ammonia or amines using conductive CPs.^{22, 23} The NDI-based SBCPs developed herein have great potential to be used as multifunctional sensors covering photoactive, chemiresistive, and chiral sensing. Their photochromic and photoswitching capabilities with redox ligands demonstrated their high potential for use in optoelectronics.

On page 12

To our knowledge, the application of homochiral CPs/MOFs have rarely been investigated in enantiodifferentiating fluorescence sensing of chiral analytes.^{43, 44} To explore the possibility of selectively sensing of homochiral SBCP towards chiral species, we investigated the chiral sensing of homochiral AlaNDI-Zn using naproxen, a nonsteroidal anti-inflammatory drug as the target analyte. The results of the fluorescence quenching titration of AlaNDI-Zn with (*R*)- or (*S*)-naproxen are shown in **Fig. 4e–h**. After adding (*R*)-naproxen (10^{-3} M) into (*S*)-AlaNDI-Zn dispersed in ethanol, obvious fluorescence quenching was observed (**Fig. 4e**). To investigate enantiodifferentiating fluorescence sensing of chiral naproxen, the relative signal intensity ratio

(I/I_0) of homochiral AlaNDI-Zn upon exposure to different enantiomeric analytes (10^{-4} M) was plotted in **Fig. 4f**. Interestingly, (*S*)-AlaNDI-Zn exhibited selectively larger fluorescence quenching ability toward (*R*)-naproxen, while (*R*)-AlaNDI-Zn showed the opposite result. Their enantioselective fluorescence quenching was further investigated for the potential application to quantify the enantiomeric excess of the chiral analytes. The relative fluorescence intensity ratio of AlaNDI-Zn was tested under different enantiomeric excess of naproxen exposure (10^{-4} M) in ethanol medium at $\lambda_{\max} = 475$ nm (**Fig. 4g**). The relative quenching of I/I_0 was linearly correlated with the enantiomeric composition of naproxen. Therefore, the enantiomeric excess of naproxen could be easily determined by a simple fluorescence quenching detection.

The fluorescence quenching data for both (*R*)- and (*S*)-naproxen to (*S*)-AlaNDI-Zn SCBPs can be readily subjected to linear regression analysis using Stern-Volmer equation:⁴⁵

$$I_0/I = 1 + K_{sv}[Q] \quad (5)$$

I_0 is the inherent fluorescence intensity of (*S*)-AlaNDI-Zn, I is the fluorescence intensity in the presence of the quencher (*R*)- or (*S*)-naproxen, K_{sv} is the Stern-Volmer constant. Based on a linear Stern-Volmer plot, K_{sv}^R and K_{sv}^S were estimated to be 5227 and 4213 M^{-1} for (*R*)- and (*S*)-naproxen, respectively (**Fig. 4h**). Accordingly, the enantiodiscrimination of ΔK and enantioselectivity factor α were also determined to be 1014 M^{-1} and 1.24, respectively, from the following equations:

$$\Delta K = (K_{sv}^R - K_{sv}^S) \quad (6)$$

$$\alpha = K_{sv}^R/K_{sv}^S \quad (7)$$

To further verify the origin of enantioselectivity of homochiral SBCPs, the binding interactions of chiral naproxen molecules were theoretically compared (**Supplementary Figs. 11 and 12**). As a result, the (*R*)-naproxen exhibited a stronger binding interaction with (*S*)-AlaNDI-Zn than its (*R*)-counterpart with increasing surface coverage, whereas the (*S*)-naproxen showed the opposite trend, as confirmed by the experimental results (**Figs. 4f and h**). Therefore, the binding energy differences of chiral analytes at the surface are predicted to be the origin of enantioselective adsorption.

On page 16

Interestingly, the photoluminescent homochiral SBCPs has been successfully developed to explore the chiral acid naproxen by fluorescence quenching with high sensitivity and selectivity owing to the different binding energies of chiral analytes at the SBCP surface.

For Figure 4e-h

Fig. 4 | Photoluminescent sensing using AlaNDI-Zn SBCPs ... e, Quenching of the fluorescence intensity of (S)-AlaNDI-Zn under (R)-naproxen (10^{-3} M) in ethanol medium. f, Relative fluorescence intensities of AlaNDI-Zn under different chiralities of naproxen (10^{-4} M) in ethanol medium at $\lambda_{\text{max}} = 475 \text{ nm}$, e.g., R-S means ((R)-AlaNDI-Zn)-(S)-naproxen). g, Relative fluorescence intensities of AlaNDI-Zn under different enantiomeric excesses of naproxen (10^{-4} M) in ethanol medium at $\lambda_{\text{max}} = 475 \text{ nm}$. h, Linear Stern-Volmer plots for the enantiodifferentiating fluorescence sensing of (S)-AlaNDI-Zn for (S)- and (R)-naproxen.

For Supplementary Figure 11

Supplementary Figure 11 | Binding configurations of chiral naproxen on homochiral AlaNDI-Zn SBCP surfaces. From top to bottom panel, the configurations of (*R*)- or (*S*)-naproxen adsorption on (*R*)-AlaNDI-Zn surface (*e.g.*, *R-R* or *R-S*), and (*R*)- or (*S*)-naproxene adsorption on (*S*)-AlaNDI-Zn surface (*e.g.*, *S-R* or *S-S*) are presented. The carbon, hydrogen, oxygen, nitrogen and zinc atoms of AlaNDI-Zn are colored in gray, white, red, blue and navy blue, respectively. The naproxen molecules adsorbed on the surface are represented in CPK style (*i.e.*, C : black, O : red, H : white).

For Supplementary Figure 12

Supplementary Figure 12 | Binding energy (ΔE_{bind}) comparison of chiral naproxen on homochiral AlaNDI-Zn surfaces. (a) (*R*)- or (*S*)-naproxen adsorption on (*R*)-AlaNDI-Zn surface. (e.g., *R-R* or *R-S*) (b) (*R*)- or (*S*)-naproxen adsorption on (*S*)-AlaNDI-Zn surface (e.g., *S-R* or *S-S*).

In Reference (added new references)

43. Chandrasekhar, P., Mukhopadhyay, A., Savitha, G. & Moorthy, J. N. Remarkably selective and enantiodifferentiating sensing of histidine by a fluorescent homochiral zn-mof based on pyrene-tetralactic acid. *Chem. Sci.* **7**, 3085-3091 (2016).
44. Wanderley, M. M., Wang, C., Wu, C.-D. & Lin, W. A chiral porous metal-organic framework for highly sensitive and enantioselective fluorescence sensing of amino alcohols. *J. Am. Chem. Soc.* **134**, 9050-9053 (2012).
45. Mei, X. & Wolf, C. Enantioselective sensing of chiral carboxylic acids. *J. Am. Chem. Soc.* **126**, 14736-14737 (2004).

2) In the introduction, the authors discussed the challenges of fabricating heterochiral structures. Could the authors elaborate what design considerations were taken to yield the heterochiral SBCP reported in this work?

Response: Thank you for your valuable comments. In general, it has been known that high fidelity homochiral or heterochiral self-sorting poses a substantial challenge due to the similarity between recognition sites of enantiomers and common conformational lability (*Chem. Rev.* **2017**, *117*, 4863-4899). As we replied to the comment 1, the idea of this work originated from our earlier study (*Adv. Mater.* **2017**, *29*, 1605828), in which the heterochiral NWs were formed by mixing the two enantiomers of PDI molecules. In this work, we wanted to explore the possibility of the formation of heterochiral coordination polymers by mixing the two enantiomeric ligands and the metal source.

3) The increased current upon exposure to electron-rich vapors is attributed to formation of the charge transfer complex. The formation of charge transfer complex may not directly contribute to increase in conductivity. Furthermore, it will be helpful to provide evidence of the charge transfer complex formation, using UV-Vis-NIR spectroscopy for instance, or through DFT calculations built upon the DFT studies already performed.

Response: Thank you for your insightful comments. According to your comments, we performed UV-vis-NIR spectroscopy analysis (**Supplementary Fig. 8**). From the UV-vis-NIR spectra, the formation of charge transfer complex was not clearly observed. Therefore, we investigated the mechanism by systematic computational studies. In order to understand the chemiresistive sensing mechanism, we carried out the periodic DFT calculations. From the DOS analysis, the reduction of bandgap was clearly observed due to the downshifting of LUMO levels (NDI π^* orbitals). Thus, we conjectured that the enhanced conductivity originated from the donor-acceptor electron transfer process. Therefore, we removed discussions about charge transfer complex and added contents about the donor-acceptor electron transfer process in computational studies. Besides, we also calculated the detection limit of the chemiresistive sensing in the revised manuscript as follows:

On page 5

Furthermore, we have used insulating SBCPs for chemiresistive sensing of electron-rich VOCs, such as methanol, ethanol, and aniline with enhanced conductivity due to the reduction of energy gap. The detection limit for electron-donating aniline reaches 16 ppm with the insulative SBCPs, which is comparable to the reported high-performance chemiresistive sensing for ammonia or amines using conductive CPs.^{22, 23}

Equation 4 is on page 11

$$(\text{Detection Limit}) = (3 \times S_b) / SL \quad (4)$$

On page 15

To theoretically elucidate the chemiresistive sensing mechanism, we investigated the electronic density of states (DOS) before and after adsorption of aniline at the (*Rac*)-AlaNDI-Zn surface by DFT calculation (**Fig. 5c-e**). Due to the adsorption of electron-rich aniline, the

peak intensity at the HOMO level corresponding to the π orbital of aniline was decreased, implying the electron donating behavior from aniline to the (*Rac*)-AlaNDI-Zn surface. Concurrently, the LUMO level corresponding to the π^* orbital of NDI ligands was downshifted, thereby resulting in the reduction of energy gap, which was demonstrated to be the origin of enhanced conductivity.

The detection limit of aniline vapor was approximately 16 ppm (according to the calculation using equation 4 and **Supplementary Fig. 13**), which was comparable to those of conductive CPs reported previously for sensing ammonia or amines.^{22, 23}

On page 16

... which leads to the reduction of the energy gap for the chemiresistive sensing. Among the tested VOCs, aniline showed the highest sensitivity with a detection limit of approximately 16 ppm, which is comparable to the reported chemiresistive detection limit of ammonia or amines using the conductive CPs.^{22, 23}

For Supplementary Figure 8

Supplementary Figure 8 | UV-Vis-NIR spectra of heterochiral AlaNDI-Zn SBCPs under exposure to hydrazine and aniline. UV-Vis-NIR spectra of (*Rac*)-AlaNDI-Zn in presence of hydrazine and aniline (0.1 M) in ethanol medium.

For Figure 5c-e

Fig. 5 | Chemiresistive sensing using heterochiral AlaNDI-Zn SBCPs and their electronic structure analysis of the heterochiral AlaNDI-Zn SBCP surface by aniline adsorption. ...
c, Electronic density of states (DOS) of aniline (red line) and the (*Rac*)-AlaNDI-Zn surface (black line) before (dotted line) and after (solid line) aniline adsorption. The red and black arrows represent the shifts in HOMO and LUMO levels, respectively. **d**, Electronic structure of HOMO level consisting of the π orbital of aniline. **e**, Electronic structure of LUMO level consisting of the π^* orbital of the NDI ligand. The carbon, hydrogen, oxygen, nitrogen and zinc atoms of (*Rac*)-AlaNDI-Zn are colored gray, white, red, blue and navy blue, respectively. The carbon, hydrogen, and nitrogen atoms of aniline are represented as large spheres colored black, cyan, and blue, respectively.

For Supplementary Figure 13

Supplementary Figure 13 | Linear plot of current changes depending on aniline concentration. Linear fitting of current changes vs. the concentration of aniline gas.

4) Regarding Figure 4, the authors claim that the detection limit to N_2H_4 is 10^{-7} M. Typically, the detection limit is not equal to the lowest concentration tested. Instead, the detection limit can be extrapolated from the linear calibration curve when the signal equals three times the root-mean-square deviation of the noise level.

Response: Thank you for your valuable comments. According to your comments, we performed additional experiments and calculated the detection limit as 3.2×10^{-6} M. we changed the content on the detection limit of PL sensing in the revised manuscript.

On page 5

In addition, the SBCPs have been applied to photoluminescence (PL) sensing of a trace amount of the harmful chemical hydrazine with the detection limit of 3.2×10^{-6} M, which is comparable to the previous report.³⁰

On page 11

PXRD showed that (*Rac*)-AlaNDI-Zn can tolerate a concentration of hydrazine in ethanol up to 0.1 M (Supplementary Fig. 7), which was also confirmed by UV-Vis-NIR spectra (Supplementary Fig. 8). The high sensitivity of (*Rac*)-AlaNDI-Zn toward lower concentration of toxic hydrazine was obtained in Fig. 4a and b. The detection limit of (*Rac*)-AlaNDI-Zn for hydrazine in PL sensing was estimated by the following equation:

$$(\text{Detection Limit}) = (3 \times S_b) / SL \quad (4)$$

where S_b is the standard deviation value of the measured signal of blank samples (the intensity of *(Rac)*-AlaNDI-Zn in ethanol medium), and SL means a slope of linearly fitted sensing results. The estimated detection limit was 3.2×10^{-6} M with a correlation coefficient $R = 0.99382$, demonstrating our *(Rac)*-AlaNDI-Zn could detect a trace amount of hydrazine by fluorescence quenching, which was comparable to that of the previous reported PL sensing of hydrazine by NDI-MOF (Fig. 4c).³⁰

On page 16

These SBCPs were successfully used for PL sensing of toxic hydrazine with a detection limit as low as 3.2×10^{-6} M.

For Figure 4c

Fig. 4 | Photoluminescent sensing using AlaNDI-Zn SBCPs ... (c) Linear fitting of intensity change vs. the concentration of hydrazine at $\lambda_{\text{max}} = 475$ nm.

Reviewer #2

Comments:

1) The manuscript entitled ‘Chiral Self-Discriminated Multifunctional Supramolecular Biocoordination Polymers (SBCPs) and Their Applications in Sensors’ by Shang et al. deals with biocoordination polymer of Naphthalenediimide (NDI) with amino acid functionalities. NDI with alanine-termini (D/L) was coordinated with zinc ion that forms homo, heterochiral, and also enantiomeric derivative with mixed R- and S-ligands. Such a property has been well studied in the literature and many of the original references are not cited. The synthesis was carried out following previous literature. They successfully formed single crystal structures of the different chiral AlaNDI-Zn derivatives. The multifunctional property of the SBCPs has been highlighted in this manuscript.

Response: Thank you for your comments. A chiral self-sorting phenomenon has been studied as a tool for the formation of discrete complex structures (*Chem. Rev.* **2017**, *117*, 4863-4899). Although some works have been reported in the chiral self-discrimination of CPs/MOFs (*Chem. Comm.* **2014**, *50*, 13829-13832., *Chem. Mater.* **2007**, *19*, 5083-5089., and *Inorg. Chem.* **2006**, *45*, 650-659.), the chiral self-discrimination of NDI-based BCPs with amino acids as the building blocks is still very rare. Although we synthesized our ligands according to the reported paper (*Carbohydr. Res.* **2005**, *340*, 1413-1418.), however, in terms of the synthesis of the BCPs, both homochiral and heterochiral BCPs are new materials.

2) This work includes measuring number of material properties and completely lacks novelty and direction required for a high impact journal. As the author claimed four main properties – photochromic, photoluminescent, photoconductive, and chemiresistive, but none of them are correlated. It would be better to focus on a specific or related properties and functionalities. All the application aspects are well reported in previous literatures.

Response: We do not agree to the reviewer’s opinion that our work completely lacks novelty and direction. It is regrettable that the reviewer did not understand the essence of our paper probably because we were not able to pass on the key point well in the manuscript. As we mentioned in our manuscript, we focused on the multifunctionality of our materials which can be used for photodetector, photoswitching, photoluminescence sensor, and chemiresistive sensor. Your comment describing that all the application aspects are well reported in previous literatures might come from misunderstanding. Most of the previously reported MOF sensors could not provide such multifunctionality in one material. Achieving such multifunctionality

in one material is really challenging and would not only greatly extend the application areas but also simplify the device fabrication procedures for many applications, such as its use in multisensors.

We have clearly demonstrated the novelty of our work in our manuscript as follows: (1) we synthesized both homochiral and heterochiral novel BCPs and found the chiral self-sorting phenomenon. (2) Even though few NDI-based CPs/MOFs used for photoluminescent sensing were reported, our system showed much higher sensitivity toward the target analyte, i.e., hydrazine, even from the rather insulating frameworks. (3) The research on the photoswitching of CPs/MOFs is still in its infancy. We firstly used redox ligand NDI-based BCPs for photoswitching properties, instead of previously reported structural or geometrical changes of the ligands upon light irradiation. Besides, for the first time, we quantitatively demonstrated the photodetecting performance using various figure of merit parameters, such as EQE, R , P , and D^* . (4) Even though our BCP was an insulator, it could be used to selectively sense electron-donating molecules such as alcohols and aniline. Furthermore, the detection limit of aniline could reach an unprecedented level of 16 ppm. In comparison with previously reported chemiresistive sensing studies using conductive MOFs, our work has novelty in that our results clearly demonstrated an efficient methodology to use insulative CPs/MOFs for chemiresistive sensing.

For the photochromic and photoswitching applications, we have already related them to each other. We carried out the UV-vis and ESR spectral analyses and revealed their mechanisms, which are related to the generation of radicals during the light irradiation. Besides, based on the PL sensing experiments using electron-donating hydrazine as the quencher, we found that the chemiresistive sensing using electron-donating materials may also be possible. Therefore, we carried out the subsequent chemiresistive sensing using aniline and alcohols as the analytes. Judging from these aspects, we carefully think that the reviewer's comment in this part was made by misunderstanding. We truly hope that this extensive revision with additional studies can be helpful for the clear understanding of our manuscript.

3) The homo- vs. heterochirality has not been discussed in depth in the manuscript, and also the author failed to connect the chirality to the functionality and not discussed with reference

to known and similar studies from the literature, that would give the reader good perspective of the work.

Response: Thank you very much for your valuable and constructive comments. For the homo- vs heterochirality, we performed the DFT calculation to explain why chiral self-sorting occurred. From the formation energy calculation, the thermodynamic preference of heterochiral SBCPs was theoretically verified. In order to better connect the chirality to the functionality, we carried out the chiral PL sensing experiment using chiral naproxen as the analyte. Also, we investigated the binding interactions of chiral naproxen molecules at the homochiral SBCP surfaces by DFT calculation in order to identify the origin of enantioselectivity.

On page 5

Interestingly, our homochiral SBCP can enantioselectively detect chiral naproxen by fluorescence quenching with high sensitivity because of their different binding strengths.

On page 7

For a better understanding of the formation of chiral self-discriminated heterochiral crystals when mixing the two enantiomeric ligands, the density functional theory (DFT) calculation of the formation energy (ΔE_f) was carried out using the following equation,

$$\Delta E_f = E_{(Rac)} - (E_{(R)} + E_{(S)})/2 \quad (1)$$

where $E_{(Rac)}$, $E_{(R)}$, and $E_{(S)}$ represent the total energy of unit cell for (*Rac*)-, (*R*)-, and (*S*)-AlaNDI-Zn SBCP, respectively (**Fig. 1d**). The ΔE_f was calculated to be -1.18 kcal/mol, indicating the formation of heterochiral AlaNDI-Zn is thermodynamically preferable than its homochiral counterpart in excellent accordance with the experimental observations.

On page 12

To our knowledge, the application of homochiral CPs/MOFs have rarely been investigated in enantiodifferentiating fluorescence sensing of chiral analytes.^{43, 44} To explore the possibility of selectively sensing of homochiral SBCP towards chiral species, we investigated the chiral sensing of homochiral AlaNDI-Zn using naproxen, a nonsteroidal anti-inflammatory drug as the target analyte. The results of the fluorescence quenching titration of AlaNDI-Zn with (*R*)-

or (*S*)-naproxen are shown in **Fig. 4e–h**. After adding (*R*)-naproxen (10^{-3} M) into (*S*)-AlaNDI-Zn dispersed in ethanol, obvious fluorescence quenching was observed (**Fig. 4e**). To investigate enantiodifferentiating fluorescence sensing of chiral naproxen, relative signal intensity ratio (I/I_0) of homochiral AlaNDI-Zn upon exposure to different enantiomeric analytes (10^{-4} M) were plotted in **Fig. 4f**. Interestingly, (*S*)-AlaNDI-Zn exhibited selectively larger fluorescence quenching ability toward (*R*)-naproxen, while (*R*)-AlaNDI-Zn showed the opposite result. Their enantioselective fluorescence quenching was further investigated for the potential application to quantify the enantiomeric excess of the chiral analytes. The relative fluorescence intensity ratio of AlaNDI-Zn was tested under different enantiomeric excess of naproxen exposure (10^{-4} M) in ethanol medium at $\lambda_{\max} = 475$ nm (**Fig. 4g**). The relative quenching of I/I_0 was linearly correlated with the enantiomeric composition of naproxen. Therefore, the enantiomeric excess of naproxen can be easily determined by a simple fluorescence quenching detection.

The fluorescence quenching data for both (*R*)- and (*S*)-naproxen to (*S*)-AlaNDI-Zn SCBPs can be readily subjected to linear regression analysis using Stern-Volmer equation:⁴⁵

$$I_0/I = 1 + K_{sv}[Q] \quad (5)$$

I_0 is the inherent fluorescence intensity of (*S*)-AlaNDI-Zn, I is the fluorescence intensity in the presence of the quencher (*R*)- or (*S*)-naproxen, K_{sv} is the Stern-Volmer constant. Based on a linear Stern-Volmer plot, K_{sv}^R and K_{sv}^S were estimated to be 5227 and 4213 M^{-1} for (*R*)- and (*S*)-naproxen, respectively (**Fig. 4h**). Accordingly, the enantiodiscrimination of ΔK and enantioselectivity factor α were also determined to be 1014 M^{-1} and 1.24, respectively, from the following equations:

$$\Delta K = (K_{sv}^R - K_{sv}^S) \quad (6)$$

$$\alpha = K_{sv}^R / K_{sv}^S \quad (7)$$

To further verify the origin of enantioselectivity of homochiral SBCPs, the binding interactions of chiral naproxen molecules were theoretically compared (**Supplementary Figs. 11 and 12**). As a result, the (*R*)-naproxen exhibited a stronger binding interaction with (*S*)-AlaNDI-Zn than its (*R*)-counterpart with increasing surface coverage, whereas the (*S*)-naproxen showed the opposite trend, as confirmed by the experimental results (**Figs. 4f and h**). Therefore, the binding energy differences of chiral analytes at the surface are predicted to be the origin of enantioselective adsorption.

On page 16

In conclusion, we synthesized homochiral and heterochiral AlaNDI-Zn SBCPs from pure and mixed alanine-based NDI enantiomeric ligands, respectively. The selective formation of heterochiral AlaNDI-Zn was due to their thermodynamic preference with lower formation energy than their homochiral counterpart.

Interestingly, the photoluminescent homochiral SBCPs has been successfully developed to explore the chiral acid naproxen by fluorescence quenching with high sensitivity and selectivity owing to the different binding energies of chiral analytes at the SBCP surface.

For Figure 1

Fig. 1 | Crystal structures of homochiral and heterochiral AlaNDI-Zn SBCPs and their chiral self-discrimination phenomenon. ... d, Formation energy (ΔE_f) calculation results for chiral discrimination phenomena in heterochiral SBCPs. The carbon, hydrogen, oxygen, nitrogen and zinc atoms of AlaNDI-Zn SBCPs are colored in gray, white, red, blue and navy blue, respectively.

For Figure 4e-h

Fig. 4 | Photoluminescent sensing using AlaNDI-Zn SBCPs. ... e, Quenching of the fluorescence intensity of (S)-AlaNDI-Zn under (R)-naproxen (10^{-3} M) in ethanol medium. f, Relative fluorescence intensities of AlaNDI-Zn under different chiralities of naproxen (10^{-4} M) in ethanol medium at $\lambda_{\text{max}} = 475$ nm, e.g., R-S means ((R)-AlaNDI-Zn)-(S)-naproxen). g, Relative fluorescence intensities of AlaNDI-Zn under different enantiomeric excesses of naproxen (10^{-4} M) in ethanol medium at $\lambda_{\text{max}} = 475$ nm. h, Linear Stern-Volmer plots for the enantiodifferentiating fluorescence sensing of (S)-AlaNDI-Zn for (S)- and (R)-naproxen.

For Supplementary Figure 11

Supplementary Figure 11 | Binding configurations of chiral naproxen on homochiral AlaNDI-Zn SBCP surfaces. From top to bottom panel, the configurations of (*R*)- or (*S*)-naproxen adsorption on (*R*)-AlaNDI-Zn surface (*e.g.*, *R-R* or *R-S*), and (*R*)- or (*S*)-naproxene adsorption on (*S*)-AlaNDI-Zn surface (*e.g.*, *S-R* or *S-S*) are presented. The carbon, hydrogen, oxygen, nitrogen and zinc atoms of AlaNDI-Zn are colored in gray, white, red, blue and navy blue, respectively. The naproxen molecules adsorbed on the surface are represented in CPK style (*i.e.*, C : black, O : red, H : white).

For Supplementary Figure 12

Supplementary Figure 12 | Binding energy (ΔE_{bind}) comparison of chiral naproxen on homochiral AlaNDI-Zn surfaces. (a) (R)- or (S)-naproxen adsorption on (R)-AlaNDI-Zn surface. (e.g., R-R or R-S) (b) (R)- or (S)-naproxen adsorption on (S)-AlaNDI-Zn surface (e.g., S-R or S-S).

In Reference (added new references)

43. Chandrasekhar, P., Mukhopadhyay, A., Savitha, G. & Moorthy, J. N. Remarkably selective and enantiodifferentiating sensing of histidine by a fluorescent homochiral zn-mof based on pyrene-tetralactic acid. *Chem. Sci.* **7**, 3085-3091 (2016).
44. Wanderley, M. M., Wang, C., Wu, C.-D. & Lin, W. A chiral porous metal-organic framework for highly sensitive and enantioselective fluorescence sensing of amino alcohols. *J. Am. Chem. Soc.* **134**, 9050-9053 (2012).
45. Mei, X. & Wolf, C. Enantioselective sensing of chiral carboxylic acids. *J. Am. Chem. Soc.* **126**, 14736-14737 (2004).

4) Authors claimed in page-3, para-2nd: “However, the π - π interaction have not been well characterized.” is not acceptable. There are several reports on this topic. Again, in page-4, para-2nd: “.....CPs as next-generation functional materials in electronics and photonics.” – the statement, however, was not justified in their work. They reported a simple photo-current generation which is well known for NDI derivatives, as they have electro-potential similar to chlorophyll. The DFT calculation is also not very informative – for these types of systems already the PET mechanism is well known. If the author can specifically comment on the chirality-induced photocurrent differences and their applications then it will be interesting.

Response: Thank you for your valuable comments. Considering your comments, we deleted the sentence “however, the π - π stacking interactions in metal-organic supramolecular networks have not been well characterized”. For the comment that our work did not justify CPs as next-

generation functional materials in electronics and photonics, we do not agree with this point. We did substantial experiments on the photoswitching and chemiresistive sensing, therefore, we believe that our work has demonstrated that BCPs we synthesized can be used in electronics and photonics. It is right that NDI molecules are well known in the photo-current generation under irradiation of light. However, for NDI-based homochiral and heterochiral CPs/MOFs, it is still rare. Besides, the research on the photoswitching of CPs/MOFs is still in its infancy. To our knowledge, we firstly quantitatively demonstrated the redox ligand-based CPs for photodetectors using various figure of merit parameters, such as EQE, P , R and D^* . For the chirality-induced photocurrent difference, we failed to get the reasonable result from CPL sensing, therefore we conducted additional research on the chiral PL sensing using homochiral SBCPs, as we discussed above.

5) Moreover, in the experimental section, they reported the activation time for photo-current is about 1 hour, which is a slow as light-induced experiments are very fast. It seems that material is photo-resistive, that's why they mentioned it as photo-sensitive. I am more curious to know what happened before getting a single peak in EPR. The time-lag has not been elucidated. The author mentioned different molecular packing in (S/R)-AlaNDI-Zn and (Rac)-AlaNDI-Zn, but the information was not extended to the different properties. The last two sections, PL sensing and chemiresistive – in my opinion, are not required. There is no significant correlation with other parts of the report.

Response: Thank you for your valuable comments. In terms of the activation time for photo-current, we think the reviewer misunderstood the content of the manuscript. The rise time for the photocurrent in the photoswitching experiment is around 6 min, not 1 h (**Fig. 3e**). In addition, as the reviewer was curious to know what happened before getting a single peak in EPR, we performed time-dependent experiments. From **Supplementary Fig. 6**, it is obvious that the intensity of the signals was gradually enhanced with increasing time. For the photoswitching experiment, we used only one single crystal, whereas we used a batch of multiple crystals for the photochromic experiment. As it is difficult to separate only one crystal for the ESR experiment, we think it is hard to quantitatively correlate these two properties. In order to test the chiral application of the homochiral BCPs, we carried out chiral PL sensing experiments, which was discussed above. As we rationalized in our paper, we want to demonstrate that our materials have multifunctional properties, therefore, we think the PL

sensing and chemiresistive sensing are necessary for our work. As described in the response to the comment 2, most of the previously reported MOF sensors could not provide such multifunctionality in one material. Achieving such multifunctionality in one material is really challenging and would not only greatly extend the application areas but also simplify the device fabrication procedures for many applications, such as its use in multisensors.

On page 9

With increasing time, the intensity of signals in ESR was gradually enhanced within 1 h, indicating more radicals were formed accordingly (**Supplementary Fig. 6**).

For Supplementary Figure 6

Supplementary Figure 6 | ESR spectra of heterochiral AlaNDI-Zn SBCPs depending on the UV light exposure time. ESR results of (*Rac*)-AlaNDI-Zn upon UV light irradiation depending on the exposure time.

Reviewer #3

Comments:

1) Hak Oh et al report on the use of supramolecular coordination polymers based on the coordination of Zn(II) ions with electroactive naphthalene diimide (NPDI) molecules functionalized with chiral alanine residues. This coordination polymers are then used to test different properties like photoconductivity, photoluminescence or chemiresistive sensing. After reading the paper it is hard to understand why the authors give so much importance to the chirality of the polymers as its influence over the properties highlighted is not explored. In my opinion, the use of chiral analytes for their experiments on sensing—either by quenching of the luminescence or electrical signal transduction— would have been substantially more interested than those presented in the work. In the current state, chirality appears of vague added value rather than reinforcing the interest of the work. Even more so, when chiroptical sensing with similar systems has been recently reported by the authors (Adv. Mater. 2017, 29 (21), 1605828). It is hard to understand why the author give this so much importance in the introduction and their discussion if chirality is not studied in any way. I don't recognise substantial novelty of broad interest and would recommend publication somewhere else.

Response: Thank you very much for your valuable comments. We agree with you in that we gave too much emphasis on the importance of chirality without doing the chiral applications. In order to explore the contribution of chirality to photoswitchability and chemiresistivity, we initially tried the circularly polarized light (CPL) detection and chiral chemiresistive sensing using homochiral BCPs, but unfortunately we failed to get the meaningful data regarding the effects of chirality on these properties, particularly due to the low dissymmetry factor in the CPL detection. This was the reason why we rather focused on the chiral self-discrimination of heterochiral crystals and their use in multifunctional sensors, since these properties themselves have rarely been reported. According to your valuable comments, we additionally performed chiral photoluminescent sensing of a chiral analyte naproxen, which is a nonsteroidal anti-inflammatory drug of the propionic acid class that relieves pain, fever, swelling, and stiffness, using homochiral BCPs, in order to investigate the effects of chirality on sensing behaviors. We also compared the binding interactions of chiral naproxen molecules at the homochiral BCP surfaces by DFT calculation to identify the origin of enantioselectivity. From these additional experiments and theoretical calculations, we obtained very interesting results on the

enantioselective chemical detection using photoluminescence quenching phenomena and revised the manuscript as follows:

On page 5

Interestingly, our homochiral SBCP can enantioselectively detect chiral naproxen by fluorescence quenching with high sensitivity because of their different binding strengths.

On page 12

To our knowledge, the application of homochiral CPs/MOFs have rarely been investigated in enantiodifferentiating fluorescence sensing of chiral analytes.^{43, 44} To explore the possibility of selectively sensing of homochiral SBCP towards chiral species, we investigated the chiral sensing of homochiral AlaNDI-Zn using naproxen, a nonsteroidal anti-inflammatory drug as the target analyte. The results of the fluorescence quenching titration of AlaNDI-Zn with (*R*)- or (*S*)-naproxen are shown in **Fig. 4e–h**. After adding (*R*)-naproxen (10^{-3} M) into (*S*)-AlaNDI-Zn dispersed in ethanol, obvious fluorescence quenching was observed (**Fig. 4e**). To investigate enantiodifferentiating fluorescence sensing of chiral naproxen, relative signal intensity ratio (I/I_0) of homochiral AlaNDI-Zn upon exposure to different enantiomeric analytes (10^{-4} M) were plotted in **Fig. 4f**. Interestingly, (*S*)-AlaNDI-Zn exhibited selectively larger fluorescence quenching ability toward (*R*)-naproxen, while (*R*)-AlaNDI-Zn showed the opposite result. Their enantioselective fluorescence quenching was further investigated for the potential application to quantify the enantiomeric excess of the chiral analytes. The relative fluorescence intensity ratio of AlaNDI-Zn was tested under different enantiomeric excess of naproxen exposure (10^{-4} M) in ethanol medium at $\lambda_{\max} = 475$ nm (**Fig. 4g**). The relative quenching of I/I_0 was linearly correlated with the enantiomeric composition of naproxen. Therefore, the enantiomeric excess of naproxen can be easily determined by a simple fluorescence quenching detection.

The fluorescence quenching data for both (*R*)- and (*S*)-naproxen to (*S*)-AlaNDI-Zn SCBPs can be readily subjected to linear regression analysis using Stern-Volmer equation:⁴⁵

$$I_0/I = 1 + K_{sv}[Q]$$

(5)

I_0 is the inherent fluorescence intensity of (*S*)-AlaNDI-Zn, I is the fluorescence intensity in the presence of the quencher (*R*)- or (*S*)-naproxen, K_{SV} is the Stern-Volmer constant. Based on a linear Stern-Volmer plot, K_{SV}^R and K_{SV}^S were estimated to be 5227 and 4213 M⁻¹ for (*R*)- and (*S*)-naproxen, respectively (**Fig. 4h**). Accordingly, the enantiodiscrimination of ΔK and enantioselectivity factor α were also determined to be 1014 M⁻¹ and 1.24, respectively, from the following equations:

$$\Delta K = (K_{SV}^R - K_{SV}^S) \quad (6)$$

$$\alpha = K_{SV}^R / K_{SV}^S \quad (7)$$

To further verify the origin of enantioselectivity of homochiral SBCPs, the binding interactions of chiral naproxen molecules were theoretically compared (**Supplementary Figs. 11 and 12**). As a result, the (*R*)-naproxen exhibited a stronger binding interaction with (*S*)-AlaNDI-Zn than its (*R*)-counterpart with increasing surface coverage, whereas the (*S*)-naproxen showed the opposite trend, as confirmed by the experimental results (**Figs. 4f and h**). Therefore, the binding energy differences of chiral analytes at the surface are predicted to be the origin of enantioselective adsorption.

On page 16

Interestingly, the photoluminescent homochiral SBCPs has been successfully developed to explore the chiral acid naproxen by fluorescence quenching with high sensitivity and selectivity owing to the different binding energies of chiral analytes at the SBCP surface.

For Figure 4e-h

Fig. 4 | Photoluminescent sensing using AlaNDI-Zn SBCPs ... e, Quenching of the fluorescence intensity of (S)-AlaNDI-Zn under (R)-naproxen (10^{-3} M) in ethanol medium. f, Relative fluorescence intensities of AlaNDI-Zn under different chiralities of naproxen (10^{-4} M) in ethanol medium at $\lambda_{\text{max}} = 475$ nm, e.g., R-S means ((R)-AlaNDI-Zn)-(S)-naproxen). g, Relative fluorescence intensities of AlaNDI-Zn under different enantiomeric excesses of naproxen (10^{-4} M) in ethanol medium at $\lambda_{\text{max}} = 475$ nm. h, Linear Stern-Volmer plots for the enantiodifferentiating fluorescence sensing of (S)-AlaNDI-Zn for (S)- and (R)-naproxen.

For Supplementary Figure 11

Supplementary Figure 11 | Binding configurations of chiral naproxen on homochiral AlaNDI-Zn SBCP surfaces. From top to bottom panel, the configurations of (*R*)- or (*S*)-naproxen adsorption on (*R*)-AlaNDI-Zn surface (*e.g.*, *R-R* or *R-S*), and (*R*)- or (*S*)-naproxene adsorption on (*S*)-AlaNDI-Zn surface (*e.g.*, *S-R* or *S-S*) are presented. The carbon, hydrogen, oxygen, nitrogen and zinc atoms of AlaNDI-Zn are colored in gray, white, red, blue and navy blue, respectively. The naproxen molecules adsorbed on the surface are represented in CPK style (*i.e.*, C : black, O : red, H : white).

For Supplementary Figure 12

Supplementary Figure 12 | Binding energy (ΔE_{bind}) comparison of chiral naproxen on homochiral AlaNDI-Zn surfaces. (a) (R)- or (S)-naproxen adsorption on (R)-AlaNDI-Zn surface. (e.g., R-R or R-S) (b) (R)- or (S)-naproxen adsorption on (S)-AlaNDI-Zn surface (e.g., S-R or S-S).

In Reference (added new references)

43. Chandrasekhar, P., Mukhopadhyay, A., Savitha, G. & Moorthy, J. N. Remarkably selective and enantiodifferentiating sensing of histidine by a fluorescent homochiral zn-mof based on pyrene-tetralactic acid. *Chem. Sci.* **7**, 3085-3091 (2016).
44. Wanderley, M. M., Wang, C., Wu, C.-D. & Lin, W. A chiral porous metal-organic framework for highly sensitive and enantioselective fluorescence sensing of amino alcohols. *J. Am. Chem. Soc.* **134**, 9050-9053 (2012).
45. Mei, X. & Wolf, C. Enantioselective sensing of chiral carboxylic acids. *J. Am. Chem. Soc.* **126**, 14736-14737 (2004).

2) Photoconductivity of NDI molecules has been already reported (Appl. Phys. Lett. 103, 053301 (2013) & ACS Appl. Mater. Interfaces, 2014, 6 (15), pp 12295–12301). As for the sensing based on photoluminescence quenching there are plenty of reports (some are referenced by the authors) and this case seems to be more relevant because the use of a toxic analyte like hydrazine. Still the CP is not porous and it is unclear to me how the sensing operates. There are no pores in the structure to accommodate the guest to induce electron transfer that might be instead limited to the surface. In fact, the authors compare their result to a porous system NDI-MOF (ref 33) and claim that works better but it is difficult to understand why. Also, supplementary Fig 6 shows substantial amorphization of the solid in contact with hydrazine that might be the actual responsible for the observed quenching of the signal (interaction with NDPI molecules in solution upon progressive degradation of the solid). My

concerns on the reported chemiresistive sensing are also based on the comparison with previous reports of similar experiments based on porous MOFs by Dinca et al. (ref 25 and 26). Again, it is hard to understand how the guest will interact with their solid beyond mere charge transfer at a superficial level. The authors do not comment on the stability of their material during the experiments (there is no data that helps evaluating this point) and it is difficult to rule out if the behaviour might be linked to NPDI molecules in solution.

Response: Thank you very much for your valuable and constructive comments. It is right that photoconductivity of NDI molecules has frequently been reported. However, these references concentrate on the direct applications of semiconducting small molecules. To the best of our knowledge, however, for the photoconductivity of NDI-based homochiral and heterochiral CPs/MOFs, it is still very rare. Besides, semiconducting small molecules and CPs/MOFs are totally different research field because CPs/MOFs are normally insulators. Although some references on NDI-based CPs/MOFs to sense electron-donating analyte hydrazine have been reported, the research on this field is still not well studied. In order to understand the mechanism of chemiresistive and photoluminescence sensing, we performed the periodic DFT calculations. As the reviewer noted, CP is not porous, so the electron transfer is expected to occur mainly on the surface. The intermolecular interaction between solid and guest analytes can be explained by HOMO/LUMO formalism, which involves a donor-acceptor electron transfer process. For chemiresistive sensing, the electronic density of states (DOS) analysis revealed that the LUMO level consisting of NDI π^* orbital was downshifted after adsorption of aniline, resulting in the reduction of bandgap, which provides an evidence for enhanced conductivity (**Fig. 5c-e**). For photoluminescence sensing, the electron transfer from the HOMO level of electron rich hydrazine to that of SBCP was energetically favorable (**Fig. 4d**). Moreover, we found that the charge transfer amount linearly increased with increasing surface coverage of hydrazine, which was consistent with experimental trends as shown in **Fig. 4c**. Consequently, we judged that the electron transfer process at the surface is sufficiently sensible, beyond the superficial level. We don't think the quenching is caused by amorphization because amorphization occurred only at a very high concentration of hydrazine, that is, 1 M and 0.1 M. From **supplementary Fig. 7**, the crystallinity of the CP is retained from 10^{-7} M to 10^{-2} M, therefore, the amorphization of the solid in contact with hydrazine can be excluded. Compared with previously reported conductive MOFs in chemiresistive sensing (*J. Am. Chem. Soc.* **2015**,

137, 13780-13783 and *Angew. Chem. Int. Ed.* **2015**, *54*, 4349-4352), our work firstly report NDI-based insulative CPs that can be used for sensing aniline with a detection limit of 16 ppm. For the stability of the material during chemiresistive sensing, we performed the PXRD analysis in **Fig. 5g**, which shows that our materials are stable during the measurement.

On page 7

The absorption peaks of the homochiral and heterochiral AlaNDI-Zn SBCPs showed a strong absorption band at 400 nm, which can be assigned to the metal-ligand charge transfer (MLCT) transition based on DFT calculations because the highest occupied molecular orbital (HOMO) consists of Zn d_{π} orbitals and the lowest unoccupied molecular orbital (LUMO) consists of ligand π^* orbitals (**Supplementary Fig. 2**). The bandgap (E_G) of (*R*)-, (*S*)-, and (*Rac*)-AlaNDI-Zn exhibited no significant differences (*i.e.*, 1.58, 1.53, and 1.60 eV, respectively).

On page 12

Based on the DFT calculations, the HOMO and LUMO energy levels of (*Rac*)-AlaNDI-Zn surface (**Fig. 4d**) indicated that electron transfer from hydrazine to (*Rac*)-AlaNDI-Zn surface was energetically favorable upon light illumination for PL sensing because the HOMO energy (-5.66 eV) of (*Rac*)-AlaNDI-Zn surface is lower than that of hydrazine (-5.47 eV).

In addition, the charge transfer amounts from hydrazine to (*Rac*)-AlaNDI-Zn surface tended to linearly increase with increasing surface coverage (**Supplementary Fig. 10a**), which was similar to experimental results (**Fig. 4c**). Intriguingly, both the binding energy (ΔE_{bind}) and differential binding energy ($\Delta\Delta E_{bind}$) exhibited a general increasing trend in a stable direction (**Supplementary Fig. 10b**). This result implied an increased stability by the formation of hydrogen bonds (HB) between adjacent hydrazine molecules (**Supplementary Fig. 10c**). As a result, the coverage-dependent HB formation at the surface was expected to cause the sensible level of electron transfer.

On page 15

To theoretically elucidate the chemiresistive sensing mechanism, we investigated the electronic density of states (DOS) before and after adsorption of aniline at the (*Rac*)-AlaNDI-

Zn surface by DFT calculation (Fig. 5c-e). Due to the adsorption of electron-rich aniline, the peak intensity at the HOMO level corresponding to the π orbital of aniline was decreased, implying the electron donating behavior from aniline to the (*Rac*)-AlaNDI-Zn surface. Concurrently, the LUMO level corresponding to the π^* orbital of NDI ligands was downshifted, thereby resulting in the reduction of energy gap, which was demonstrated to be the origin of enhanced conductivity.

On page 15

To further demonstrate the stability of the devices under exposure to the aniline vapor, PXRD analysis was conducted (Fig. 5g). The similar signals before and after exposure to the saturated aniline vapors suggest that the crystallinity of (*Rac*)-AlaNDI-Zn was not affected by the aniline analyte.

For Figure 4d

Fig. 4 | Photoluminescent sensing using AlaNDI-Zn SBCPs ... (d) Photoluminescent sensing mechanism based on HOMO/LUMO formalism of heterochiral AlaNDI-Zn SBCP surface and hydrazine. The carbon, hydrogen, oxygen, nitrogen and zinc atoms of AlaNDI-Zn SBCPs are colored in gray, white, red, blue and navy blue, respectively.

For Figure 5g

Fig. 5 | Chemiresistive sensing using heterochiral AlaNDI-Zn SBCPs and their electronic structure analysis of the heterochiral AlaNDI-Zn SBCP surface by aniline adsorption. ...

g, PXRD results of heterochiral AlaNDI-Zn crystals after exposure to saturated aniline and ethanol gas for 30 mins.

For Supplementary Figure 10

Supplementary Figure 10 | Charge transfer and binding interactions of hydrazine on heterochiral AlaNDI-Zn SBCP surface. (a) The accumulated negative charge of (Rac)-

AlaNDI-Zn surface with increasing number of adsorbed hydrazine molecules obtained by Mulliken population analysis. (b) The binding energy (ΔE_{bind}) and differential binding energy ($\Delta\Delta E_{bind}$) with increasing number of adsorbed hydrazine molecules. (c) Binding configurations of hydrazine molecules adsorbed on (*Rac*)-AlaNDI-Zn surface. The gray colored region represents the Connolly surface of (*Rac*)-AlaNDI-Zn for the clear view of the adsorbed configurations of hydrazine. The red dotted ovals represent the hydrogen bond formation between adjacent hydrazine molecules.

3) Also important, the properties reported are intimately linked to the use of NPDI connectors assembled into a 3D solid. Still, the authors invoke DFT calculations based on simple clusters rather than modelling the periodic structures. This is key to establish an origin for the observed phenomena but I think VASP methodologies with periodic boundary conditions would be more adequate to establish clear correlations with the electronics structure of the solid rather than the HOMO/LUMO formalism used, which appears more adequate for isolated molecules. Their approach is probably mistreating the effect that π - π stacking between neighbouring NPDI molecules (clearly observed from SCXR data) might have over the electronic structure. This is key to the work as it is expected to dictate both photoconductivity and chemiresistive sensing.

Response: We thank the reviewer for the insightful comments on the fact that simple cluster calculations would not be adequate for explaining the observed phenomena, especially for the sensing mechanism. As the reviewer noted, it is reasonable to interpret with periodic models since the π - π stacking effect could not be excluded in our case. According to the valuable comments, we performed the periodic DFT calculations using DMol³ program, which has the same level of accuracy compared with VASP. As stated above, we tried to theoretically elucidate the PL quenching and chemiresistive sensing mechanisms by investigating the electronic structures of the periodic systems (**Fig. 4d and Fig. 5c-e**). It was revealed that HOMO/LUMO formalism induced by the changes of electronic structure in periodic systems possibly explained the donor-acceptor electron transfer process. Accordingly, we replaced all the data including HOMO/LUMO levels obtained from cluster models to those from periodic systems and related Experimental Section in the revised manuscript as follows.

On page 7

The absorption peaks of the homochiral and heterochiral AlaNDI-Zn SBCPs showed a strong absorption band at 400 nm, which can be assigned to the metal-ligand charge transfer (MLCT) transition based on DFT calculations because the highest occupied molecular orbital (HOMO) consists of Zn d_{π} orbitals and the lowest unoccupied molecular orbital (LUMO) consists of ligand π^* orbitals (**Supplementary Fig. 2**). The bandgap (E_G) of (*R*)-, (*S*)-, and (*Rac*)-AlaNDI-Zn exhibited no significant differences (*i.e.*, 1.58, 1.53, and 1.60 eV, respectively).

On page 20

Model Systems for Calculation. The unit cell structures of homo- and heterochiral AlaNDI-Zn SBCPs, obtained from experimental XRD analysis, were optimized by DFT calculation while keeping the experimental lattice parameters. Next, the surface systems were modeled by introducing the vacuum in the (100) direction where the AlaNDI-Zn fragments were stacked. Note that to fill the dangling sites of Zn atoms, two AlaNDI-Zn ligands were added (*i.e.*, 356 atoms). By Monte Carlo simulation, the initial binding configuration in a fixed loading of analytes (*i.e.*, hydrazine, aniline, and chiral naproxen) was found by sequentially adding the analyte molecules on the bare AlaNDI-Zn surfaces. Subsequently, the optimized structures were obtained by full relaxation through DFT calculation while keeping the crystallographic positions of all atoms in the AlaNDI-Zn surfaces.

DFT Calculations. All DFT calculations were performed with DMol³ program^{47,48} with the generalized gradient approximation (GGA) with the Perdew-Burke-Ernzerhof (PBE) functional⁴⁹, in which the semi-empirical Grimme scheme⁵⁰ for the dispersion correction was included. The spin-polarized calculations were performed with the basis set of DNP 4.4. The DFT semi-core pseudopotentials (DSPPs) were applied for all model systems. The Brillouin-zone was sampled by a Monkhorst-Pack⁵¹ as Γ -point for all model systems. The convergence criteria for energy, force, and displacement were set as 1.0×10^{-5} Ha, 0.002 Ha/Å and 0.005 Å, respectively. The atomic charges were obtained from Mulliken population analysis⁵². For the density of states (DOS) analysis, the smearing width was set to be 0.05 eV.

Monte Carlo Simulations. Monte Carlo simulations with a fixed number of analytes (*i.e.*, hydrazine, aniline, and naproxen) were carried out by Sorption program⁵³ to find the preferential adsorption sites on AlaNDI-Zn surfaces. For the interaction energy parameters,

Universal forcefield⁵⁴ was employed with atomic charges obtained from Mulliken population analysis. The systems were equilibrated for 1×10^6 MC steps and analyzed for the next 1×10^6 MC steps. The non-bond interactions, including electrostatic and van der Waals forces, were estimated by Ewald summation and atom-based cutoff (a radius of 12.5 Å) scheme.

For Supplementary Figure 2

Supplementary Figure 2 | Energy level comparison of unit cells for (R)-, (S)-, and (Rac)-AlaNDI-Zn SBCPs. The carbon, hydrogen, oxygen, nitrogen and zinc atoms of (Rac)-AlaNDI-Zn are colored in gray, white, red, blue and navy blue, respectively.

For Figure 4d

Fig. 4 | Photoluminescent sensing using AlaNDI-Zn SBCPs ... (d) Photoluminescent sensing mechanism based on HOMO/LUMO formalism of heterochiral AlaNDI-Zn SBCP surface and hydrazine. The carbon, hydrogen, oxygen, nitrogen and zinc atoms of AlaNDI-Zn SBCPs are colored in gray, white, red, blue and navy blue, respectively.

For Figure 5c-e

Fig. 5 | Chemiresistive sensing using heterochiral AlaNDI-Zn SBCPs and their electronic structure analysis of the heterochiral AlaNDI-Zn SBCP surface by aniline adsorption. ...

c, Electronic density of states (DOS) of aniline (red line) and the (*Rac*)-AlaNDI-Zn surface (black line) before (dotted line) and after (solid line) aniline adsorption. The red and black arrows represent the shifts in HOMO and LUMO levels, respectively. **d**, Electronic structure of HOMO level consisting of the π orbital of aniline. **e**, Electronic structure of LUMO level consisting of the π^* orbital of the NDI ligand. The carbon, hydrogen, oxygen, nitrogen and zinc atoms of (*Rac*)-AlaNDI-Zn are colored gray, white, red, blue and navy blue, respectively. The carbon, hydrogen, and nitrogen atoms of aniline are represented as large spheres colored black, cyan, and blue, respectively.

REVIEWERS' COMMENTS:

Reviewer #1 (Remarks to the Author):

The authors have adequately addressed the critical concerns raised in the initial review. In particular, with regard to the role of chirality in the molecular design, the authors demonstrated enantioselective detection of chiral naproxen and supported the experimental data with computational studies. The experimental study is solid and convincing, although the computational binding energies hardly showed any difference between R and S enantiomers. The authors further corrected the wrong claim of charge transfer complex formation for sensing electron-rich VOCs, as supported by the computational study. This referee would like to point out that the change in band gap upon VOC adsorption should be observable through UV-Vis spectroscopy, although it is understandable that the experiment is non-trivial as it may require in operando measurements. That said, it is not absolutely necessary to demonstrate experimental evidence in this work. The authors also addressed the LOD issue raised. Therefore, the manuscript is suitable for publication in Nature Communications.

Reviewer #2 (Remarks to the Author):

I do not see much improvement in the ms of addressing the earlier comments. The only data added is the chiral sensing of a drug, that works by fluorescence quenching which is not preferred method for any sensing applications. The response is not very appreciable as well. Therefore I cannot recommend this work for publication in Nature Communications.

Reviewer #3

Only made comments to the editor, stating the revisions are satisfactory.

We would like to thank all the reviewers for their constructive comments and suggestions on the manuscript to improve our manuscript. We have carefully studied the reviewers' suggestions. We sincerely appreciate all the reviewers' time and efforts to review our manuscript. Our point-by-point responses to the reviewers' comments are addressed below.

Reviewer #1

Comments:

The authors have adequately addressed the critical concerns raised in the initial review. In particular, with regard to the role of chirality in the molecular design, the authors demonstrated enantioselective detection of chiral naproxen and supported the experimental data with computational studies. The experimental study is solid and convincing, although the computational binding energies hardly showed any difference between R and S enantiomers.

Response: Thank you for your valuable comments. As shown in **Supplementary Figs. 11 and 12**, the tendency for the calculated binding energy differences of chiral naproxens agreed with the experimental observations, but the differences were small (i.e., < 2 kcal/mol for different molecule) because the differences in binding interactions of chiral molecules are originally very small due to their similarity of molecular structure. Notably, the scale of energy differences is in similar level to the reported values in other literatures (i.e., 0.6 ~ 3.0 kcal/mol for different molecule, *Nat. Mater.* **2003**, 2, 367-374, *Angew. Chem. Int. Ed.* **2005**, 44, 7761-7764, and *Angew. Chem. Int. Ed.* **2013**, 52, 3394-3397).

The authors further corrected the wrong claim of charge transfer complex formation for sensing electron-rich VOCs, as supported by the computational study. This referee would like to point out that the change in band gap upon VOC adsorption should be observable through UV-Vis spectroscopy, although it is understandable that the experiment is non-trivial as it may require in operando measurements. That said, it is not absolutely necessary to demonstrate experimental evidence in this work.

Response: We agree with the reviewer' comment that, in principle, the change of band gap upon VOC adsorption should be observable through UV-Vis spectroscopy. In our previous experimental condition, we used the analyte aniline in ethanol solution. However, no band gap difference was observed. We thought the reason might be related to the solvent effect of ethanol. For example, the interaction between CPs and aniline could become weak after the aniline was

surrounded by ethanol through hydrogen bonding. Therefore, we tried to use aniline vapors as the alternative analyte, which could enhance the interaction between aniline and CPs (**Supplementary Figure 8 in this revised manuscript**). Unfortunately, similar results were obtained, and the *in-situ* monitoring of the band gap change upon exposure to aniline was difficult in our experimental set-up. We assume the reasons as follows: 1; compared with our real-time chemiresistive sensing experiment, the UV-Vis measurement of the samples was carried out with a time interval after preparing the samples. 2; the adsorption of aniline only happened on the surface of the CPs, which means the interaction between the CPs and aniline was not very strong and possibly reversible.

For Supplementary Figure 8

Supplementary Figure 8 | UV-Vis-NIR spectra of heterochiral AlaNDI-Zn SBCPs under exposure to hydrazine and aniline. UV-Vis-NIR spectra of (*Rac*)-AlaNDI-Zn in presence of hydrazine and aniline (0.1 M) in ethanol medium and after exposure to saturated hydrazine and aniline gas.

The authors also addressed the LOD issue raised. Therefore, the manuscript is suitable for publication in Nature Communications.

Response: Thank you very much for your constructive and positive comments.

Reviewer #2

Comments:

I do not see much improvement in the ms of addressing the earlier comments. The only data added is the chiral sensing of a drug, that works by fluorescence quenching which is not preferred method for any sensing applications. The response is not very appreciable as well. Therefore I cannot recommend this work for publication in Nature Communications.

Response: Thank you for your comments. Although we respect the reviewer's comments, we cannot fully agree with all the reviewer's opinions. For example, the reviewer mentioned that we only added chiral sensing data, however, we spent several months carrying out the simulation studies to systematically explain the fundamental mechanisms in chiral self-discrimination, PL sensing, chemiresistive sensing, and chiral sensing. According to the previous reported papers (*Chem. Soc. Rev.* **2018**, *47*, 4729-4756. *Chem. Soc. Rev.* **2017**, *46*, 3242-3285.), using luminescent CPs/MOFs to sense various analytes, such as biomolecules, environmental toxins, explosives, ionic species is a very effective and powerful way for sensing applications. Besides, the PL instrumentation is a relatively cheap yet highly sensitive tool, and can be easily used for practical applications. The nonsteroidal anti-inflammatory drug naproxen has the property to relieve pain, fever, swelling, and stiffness. Therefore, we believe that our chiral PL sensing results using luminescent CPs are of great importance and can be easily expanded to other platforms for practical applications.

Having performed these extensive experimental and theoretical studies for the chiral self-discrimination, PL sensing, chemiresistive sensing, and chiral sensing, we strongly believe that our revised manuscript has been significantly improved. We have also carefully checked all the issues raised by the reviewers. We do hope the reviewer can now acknowledge our efforts for this revision.

Reviewer #3

Comment:

Only made comments to the editor, stating the revisions are satisfactory.

Response Thank you very much for your positive comments on the revised manuscript. We truly appreciate your time and efforts for reviewing our manuscript.